# Differences and similarities between human and chimpanzee neural progenitors during cerebral cortex development

Felipe Mora-Bermúdez[1†], Farhath Badsha[1†], Sabina Kanton[2†], J Gray Camp[2†], Benjamin Vernot[2], Kathrin Köhler[2], Birger Voigt[3], Keisuke Okita[4], Tomislav Maricic[2], Zhisong He[5], Robert Lachmann[6], Svante Pääbo[2*], Barbara Treutlein[1,2*], Wieland B Huttner[1*]

[1]Max Planck Institute of Molecular Cell Biology and Genetics, Dresden, Germany; [2]Max Planck Institute for Evolutionary Anthropology, Leipzig, Germany; [3]Institute of Laboratory Animals, Graduate School of Medicine, Kyoto University, Kyoto, Japan; [4]Department of Reprogramming Science, Center for iPS Cell Research and Application, Kyoto University, Kyoto, Japan; [5]CAS-MPG Partner Institute for Computational Biology, Shanghai, China; [6]Universitätsklinikum Carl Gustav Carus, Klinik und Poliklinik für Frauenheilkunde und Geburtshilfe, Technische Universität Dresden, Dresden, Germany

*For correspondence: paabo@ eva.mpg.de (SP); barbara_treutlein@eva.mpg.de (BT); huttner@mpi-cbg.de (WBH)

†These authors contributed equally to this work

Competing interests: The authors declare that no competing interests exist.

**Abstract** Human neocortex expansion likely contributed to the remarkable cognitive abilities of humans. This expansion is thought to primarily reflect differences in proliferation *versus* differentiation of neural progenitors during cortical development. Here, we have searched for such differences by analysing cerebral organoids from human and chimpanzees using immunohistofluorescence, live imaging, and single-cell transcriptomics. We find that the cytoarchitecture, cell type composition, and neurogenic gene expression programs of humans and chimpanzees are remarkably similar. Notably, however, live imaging of apical progenitor mitosis uncovered a lengthening of prometaphase-metaphase in humans compared to chimpanzees that is specific to proliferating progenitors and not observed in non-neural cells. Consistent with this, the small set of genes more highly expressed in human apical progenitors points to increased proliferative capacity, and the proportion of neurogenic basal progenitors is lower in humans. These subtle differences in cortical progenitors between humans and chimpanzees may have consequences for human neocortex evolution.

## Introduction

The expansion of the neocortex during primate evolution is thought to contribute to the higher cognitive capacity of humans compared to our closest living relatives, the great apes, and notably the chimpanzees (*Geschwind and Rakic, 2013*; *Rakic, 2009*; *Striedter, 2005*). Neocortex expansion in humans relative to chimpanzees involves an increase in the number of cortical neurons generated during fetal development (*Borrell and Reillo, 2012*; *Florio and Huttner, 2014*; *Herculano-Houzel, 2009*; *Lui et al., 2011*). This reflects primarily a greater and prolonged proliferative capacity of human neural stem and progenitor cells (NSPCs) within the germinal zones of the developing

**eLife digest** The human brain is about three times as big as the brain of our closest living relative, the chimpanzee. Moreover, a part of the brain called the cerebral cortex – which plays a key role in memory, attention, awareness and thought – contains twice as many cells in humans as the same region in chimpanzees. Networks of brain cells in the cerebral cortex also behave differently in the two species.

How these species differences arise is not clear, but it likely occurs in the earliest phases of development when brain stem and progenitor cells divide and give rise to cerebral cortex cells in the growing brain. To study the earliest stages of brain development, researchers often use human brain cells grown in the laboratory. Under the right conditions, cells collected from adult humans and other animals can be reprogrammed to behave like brain stem cells. Recently, researchers have been able to use these reprogrammed cells to make tissue that resembles the brain in petri dishes, known as brain organoids.

Mora-Bermúdez, Badsha, Kanton, Camp et al. have now analysed brain organoids grown from reprogrammed human, chimpanzee and orangutan cells. The experiments showed that the human and chimpanzee brain organoids were remarkably similar in many ways including in the mix of cell types and in how these cells were arranged.

Mora-Bermúdez et al. then used live microscopy to show that progenitor cells that form the human cerebral cortex spend around 50% more time in a stage of the cell division process called metaphase compared to the same cells from chimpanzees or orangutans. Metaphase is the part of the division process when the cell makes sure that structures called chromosomes, which carry the cell's DNA, can be separated and distributed equally between the two daughter cells. Mora-Bermúdez et al. also found that progenitor cells more likely to become neurons sooner had a shorter metaphase than progenitor cells more likely to remain proliferating as stem cells for longer. This suggests that a longer metaphase may be a feature of brain stem cells.

Further studies are now needed to find out how the length of time these progenitor cells spend in metaphase affects how chimpanzee and human brains develop; and whether this can help explain why the human brain is so much larger.

neocortex (*Lewitus et al., 2013*). Unravelling differences between human and chimpanzee NSPC behaviour is therefore a key issue, yet very little is known about such differences.

The neocortex develops from two principal germinal zones, the ventricular zone (VZ) and the sub-ventricular zone (SVZ) (*Angevine et al., 1970*). In primates developing a folded (gyrencephalic) neo-cortex, and notably in humans, an inner SVZ (iSVZ) and an outer (oSVZ) can be distinguished (*Dehay et al., 2015*; *Smart et al., 2002*). Correspondingly, the VZ and SVZ harbour the cell bodies of two principal classes of NSPCs, called apical progenitors (APs) and basal progenitors (BPs), respectively, each of which comprise several distinct NSPC types (*Borrell and Reillo, 2012*; *Götz and Huttner, 2005*; *Lui et al., 2011*; *Taverna et al., 2014*). APs (neuroepithelial cells, apical radial glia, and apical intermediate progenitors) divide at the ventricular surface, keep ventricular contact and exhibit apical cell polarity, whereas BPs (basal (or outer) radial glia and basal intermediate progenitors) lack this contact and type of cell polarity (*Taverna et al., 2014*).

Studies dissecting the switch between NSPC proliferation and differentiation have demonstrated that a central aspect of the cell division process, the orientation of the mitotic spindle, has a pivotal role, particularly in the case of APs (*Lancaster and Knoblich, 2012*; *Mora-Bermudez and Huttner, 2015*; *Mora-Bermudez et al., 2014*; *Shitamukai and Matsuzaki, 2012*). The orientation of the spindle relative to the apical-basal axis of cell polarity in mitotic apical radial glia, the major cortical neural stem cells (*Götz and Huttner, 2005*; *Kriegstein and Alvarez-Buylla, 2009*), can determine whether their division is symmetric or asymmetric, and whether it is proliferative or neurogenic, with regard to their progeny (*Lancaster and Knoblich, 2012*; *Mora-Bermudez and Huttner, 2015*; *Mora-Bermudez et al., 2014*; *Shitamukai and Matsuzaki, 2012*). Comparing spindle orientation in mitotic APs may therefore provide insight into the cell biological basis underlying the differences

between humans and chimpanzees in NSPC proliferation *versus* differentiation during neocortex development.

Protocols to generate structured cerebral tissue (cerebral organoids) from pluripotent stem cells in vitro constitute a major advance for studying neocortex development, in particular with regard to humans and non-human primates where fetal brain tissue is hard or impossible to obtain and manipulate (*Kadoshima et al., 2013*; *Lancaster and Knoblich, 2014*; *Lancaster et al., 2013*; *Mariani et al., 2015*; *Qian et al., 2016*). Human cerebral organoids form a variety of tissues that resemble specific brain regions, including the cerebral cortex, ventral forebrain, midbrain-hindbrain boundary, hippocampus, and retina. Moreover, their cerebral cortex-like regions exhibit distinct germinal zones, that is, a VZ containing APs and an SVZ containing BPs, as well as basal-most neuronal layers. Cerebral organoid APs include apical radial glia-like NSPCs that contact a ventricle-like lumen, express radial glia marker genes, undergo interkinetic nuclear migration, and divide at the apical surface, similar to their in vivo counterparts, and cerebral organoid BPs comprise both basal radial glia-like and basal intermediate progenitor-like NSPCs (*Lancaster et al., 2013*). Finally, we have previously shown by single-cell RNA sequencing that the gene expression programs controlling neocortex development in human cerebral organoids are remarkably similar to those in the developing fetal tissue (*Camp et al., 2015*). Together, these findings suggest that cerebral organoids constitute a valid system to explore potential differences in NSPC proliferation *versus* differentiation between humans and chimpanzees (*Otani et al., 2016*), in particular with regard to spindle orientation in mitotic APs.

Here, we have generated cerebral organoids from chimpanzee-derived induced pluripotent stem cells (iPSCs), and used single-cell transcriptomics, immunohistofluorescence and live imaging to compare relevant features of chimpanzee NSPCs to human NSPCs in cerebral organoids and fetal neocortex. While most NSPC characteristics are found to be similar, we show that the prometaphase-metaphase in mitotic APs is longer in humans than in chimpanzees, indicating that a fundamental difference exists in the regulation of mitosis during neocortex development between the two species. Our data also provide a resource for further studies on human and chimpanzee differences in cortical development, and demonstrate the usability of cerebral organoids as a means to be able to perform such studies.

## Results

### Chimpanzee cerebral organoids recapitulate cortex development

We generated cerebral organoids from iPSCs derived from chimpanzee fibroblasts and lymphocytes (*Figure 1A* left, *Figure 1—figure supplement 1*). These chimpanzee cerebral organoids formed complex tissue structures that resembled the developing primate brain (*Figure 1A* right), as reported previously for human cerebral organoids (*Lancaster et al., 2013*). Similar to human iPSC-derived cerebral organoids ([*Camp et al., 2015*], *Figure 1B,C* right), within the chimpanzee organoids grown for 52 days (D52), we observed cortex-like regions (*Figure 1A* right) with PAX6-positive APs (such as radial glia) residing predominantly in the apical-most zone facing a ventricular lumen (*Figure 1B* left), similar to the ventricular zone (VZ) of developing primate neocortex at an early-mid stage of neurogenesis. Consistent with this, cells immunoreactive for the deep-layer neuron marker CTIP2 were observed in the basal region of the developing cortical wall (*Figure 1B* left), corresponding to an early cortical plate. TBR2 (also known as EOMES) positive BPs (presumably mostly basal intermediate progenitors) were concentrated in a zone between the PAX6+ progenitors and the CTIP2+ neurons, corresponding to the subventricular zone (SVZ). In the context of the time-lapse live imaging of apical mitoses described below, we observed apically directed nuclear migration prior to, and basally directed nuclear migration after, mitosis, consistent with the existence of interkinetic nuclear migration. Our results suggest that chimpanzee cerebral organoids recapitulate important aspects of fetal chimpanzee brain development and allow comparisons with cerebral cortex development in human cerebral organoids and fetal neocortex.

We next compared the proportion of various NSPC types, as revealed by expression of PAX6 and/or TBR2, and neurons at a very early (D28) and a mid-neurogenic (D52-D54) stage between human and chimpanzee cerebral organoid cortices (*Figure 2*). In both species, we observed a decrease in PAX6+TBR2– apically located NSPCs (presumably proliferating APs) from D28 to D52,

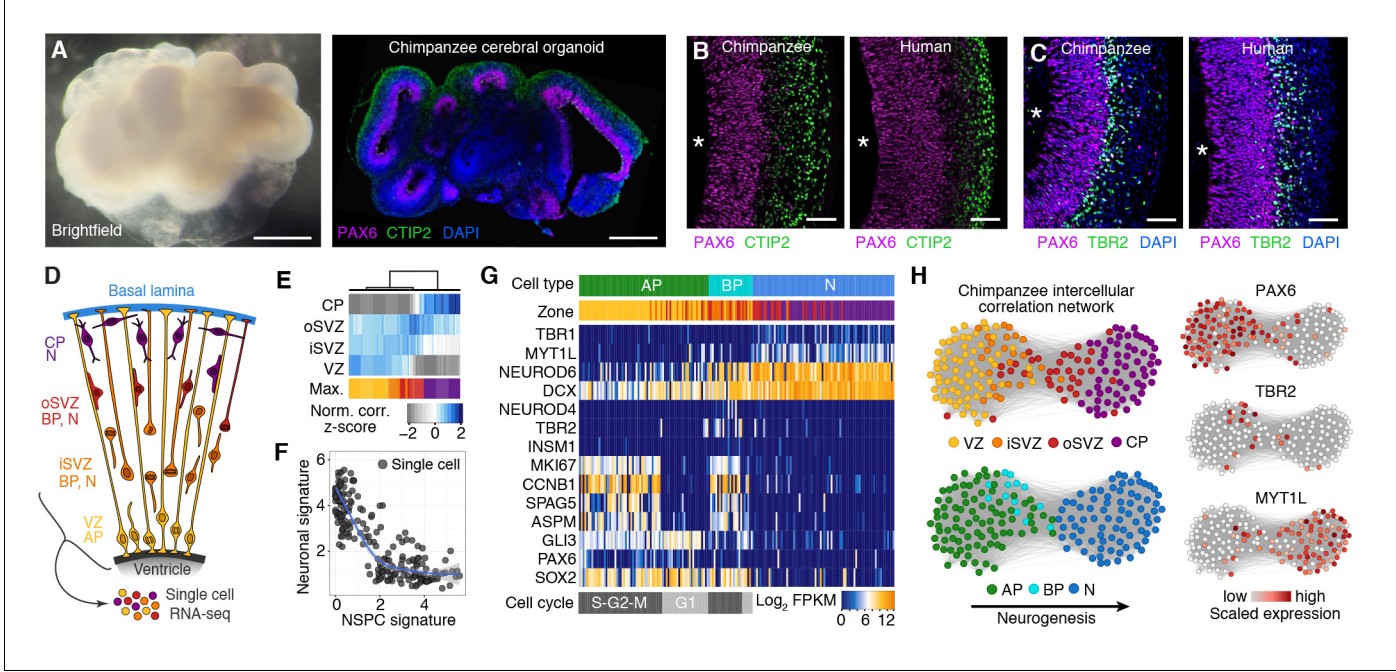

**Figure 1.** Chimpanzee cerebral organoids recapitulate cortex development. (**A**) Bright-field image showing a representative chimpanzee organoid (Sandra A, left) and a cryosection from a chimpanzee organoid (PR818-5) immunostained for PAX6 (magenta) and Ctip2 (green) combined with DAPI staining (blue) (right) at day 52. Scale bars, 200 μm. (**B, C**) Cryosections of cortical regions from chimpanzee (Sandra A) and human (SC102A-1) organoids at day 52 immunostained for PAX6 (magenta), Ctip2 (B, green) and TBR2 (C, green), without (**B**) and with (**C**) DAPI staining (blue). Asterisks, ventricular lumen; scale bars, 50 μm. (**D**) Cartoon showing NSPC types (APs, BPs) and neurons enriched in zones within the neocortex at mid-neurogenesis. CP, cortical plate; N, neuron. (**E**) Heatmap showing normalized correlation (Z-score) of single-cell transcriptomes from chimpanzee cerebral organoid cortex with bulk RNA-seq data from laser-microdissected zones (*Fietz et al., 2012*) from 13 wpc human neocortex. CP, cortical plate. (**F**) Scatterplot showing NSPC and neuronal signature scores derived from analysis of fetal cerebral cortex single-cell transcriptomes (*Figure 1—figure supplement 1*) calculated for each chimpanzee cerebral organoid cortical cell. (**G**) Heatmap showing expression of AP, BP, and neuron (N) marker genes. Each column represents a single cell, each row a gene. Cell type and maximum correlation to bulk RNA-seq data from cortical zones are shown in the top sidebar. APs and BPs were sub-classified based on G1 (light grey) or S-G2-M (dark grey) phases of the cell cycle. (**H**) Lineage network based on pairwise correlations between chimpanzee cerebral organoid cortical cells reveals a structured topology where VZ-APs connect to cortical plate (CP) neurons (N) through SVZ-BPs. Cells are coloured based on cortical zone (top left) or cell type assignment (bottom left). APs, BPs, and neurons were classified based on maximum correlation with single-cell transcriptomes from the human fetal neocortex. Expression of markers PAX6, TBR2, and MYT1L are shown to the right.

The following source data and figure supplements are available for figure 1:

**Source data 1.** Processed single-cell RNA-seq data for chimpanzee cells.

**Source data 2.** Genes describing cell populations in the chimpanzee organoids.

**Figure supplement 1.** Characterization of chimpanzee iPSCs.

**Figure supplement 2.** Deconstructing cell type composition in chimpanzee cerebral organoids using single-cell RNA-seq.

**Figure supplement 3.** Fetal human progenitor and neuronal neocortical signatures are recapitulated in chimpanzee cerebral organoids.

concomitant with an increase in PAX6+TBR2+ and PAX6–TBR2+ basally located NSPCs (presumably neurogenic BPs) (*Figure 2A,B*). Notably, whereas no significant differences were observed at D28, at D52-D54, the proportion of PAX6+TBR2+ NSPCs in the chimpanzee organoids was nearly twice that in the human organoids, and the proportion of PAX6+TBR2– NSPCs was correspondingly lower, whereas no significant difference between human and chimpanzee was observed for PAX6–TBR2+ NSPCs (*Figure 2B*). In line with what would be expected with regard to neuron production, the

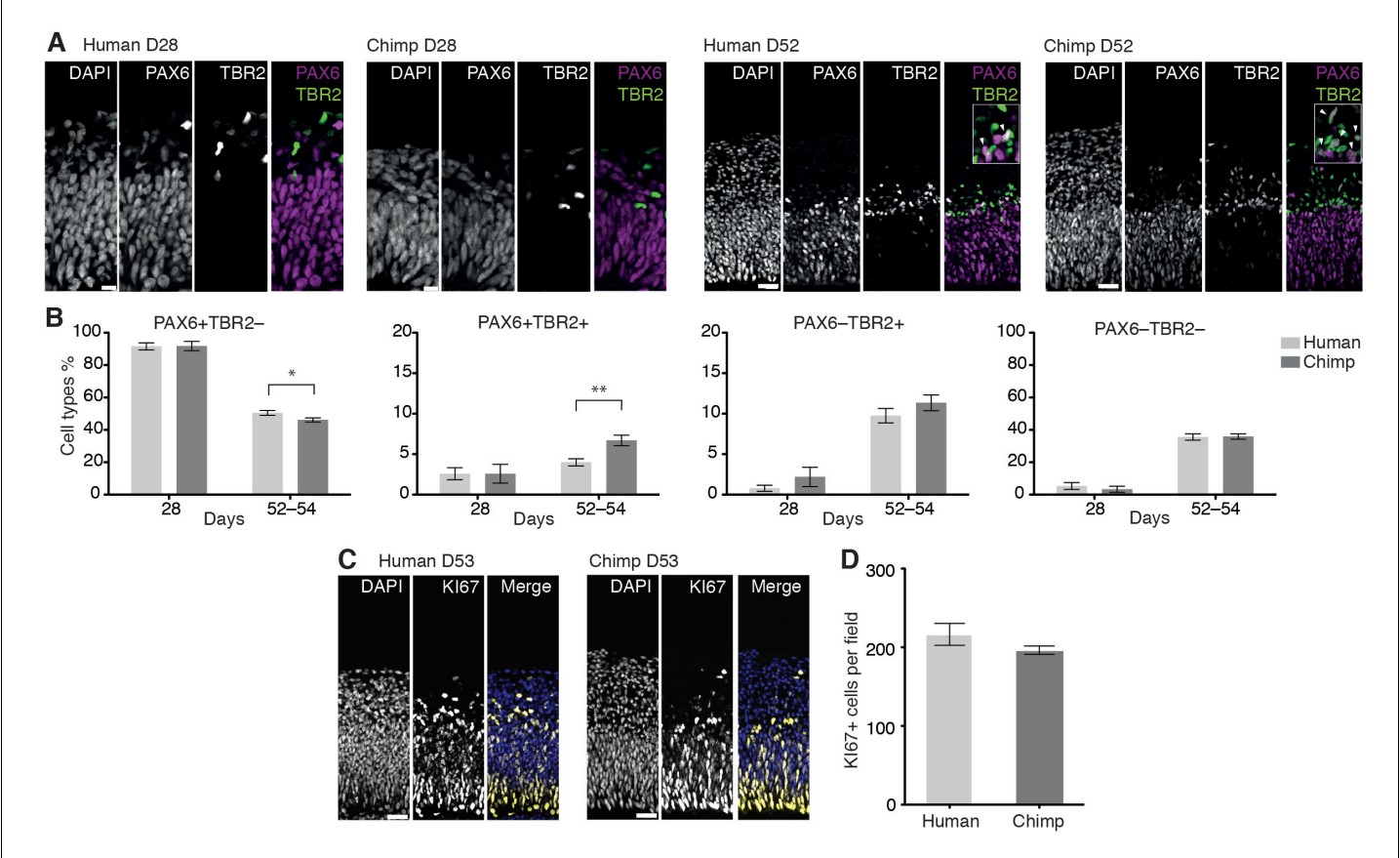

**Figure 2.** Changes in the proportion of cortical NSPC subtypes and neurons during human and chimpanzee cerebral organoid development. (**A**) Cryosections of cortical regions from human and chimpanzee organoids at day 28 and day 52 immunostained for PAX6 (magenta) and TBR2 (green) combined with DAPI staining. Scale bars; D28, 10 μm; D52, 20 μm. Insets in the D52 merge images show selected areas with PAX6+TBR2+ double-positive nuclei (arrowheads) at higher magnification. (**B**) Quantification of the percentage of PAX6+TBR2–, PAX6+TBR2+, PAX6–TBR2+ and PAX6–TBR2– cortical cells in human (light grey) and chimpanzee (dark grey) organoids at D28 (n = 5 organoids, 50 μm wide field) and D52-D54 (n = 17 organoids, 100 μm wide field). Error bars, SEM; *p<0.05, **p<0.01. (**C**) Cryosections of cortical regions from human and chimpanzee organoids at D53 immunostained for KI67 (yellow) combined with DAPI staining (blue). Scale bars, 20 μm. (**D**) Quantification of KI67+ cells in a 100 μm wide field in human and chimpanzee organoids at D52-D53 (n = 7). Error bars, SEM.

proportion of PAX6–TBR2– cells, located in the basal-most zones of the developing cortical wall, was very low at D28 but increased by D52-D54 to about a third of the total cells for both, human and chimpanzee cerebral organoids (*Figure 2B*). Immunostaining for CTIP2 corroborated the neuronal identity of these cells (data not shown).

Consistent with the observation that the total proportion of NSPCs relative to neurons was virtually identical in human and chimpanzee organoids (*Figure 2B*), the abundance of cycling cells, as revealed by KI67 immunostaining, was essentially similar (*Figure 2C,D*). We conclude that at the two stages studied, there are – with the exception of the PAX6+TBR2+ NSPCs – no major differences between human and chimpanzee cerebral cortex developing in organoid culture with regard to the types of NSPCs and their abundance, or neuron output.

## Cell composition and lineage relationships in chimpanzee cerebral organoids

To survey the cellular composition and cell type-specific transcriptomes of the chimpanzee organoids, we analysed 344 single cell transcriptomes from 7 organoids ranging in age from 45 to 80 days (*Figure 1D*, *Figure 1—source data 1*). We combined all transcriptomes and identified the genes most informative for defining cell populations by principal component analysis (PCA) (*Figure 1—*

*source data 2*). Using these genes, we used tSNE analysis to cluster cells into transcriptionally distinct groups representing cerebral cortex, hindbrain, ventral midbrain and peripheral mesenchyme (*Figure 1—figure supplement 2*). These groups are similar to those identified in human cerebral organoids (*Camp et al., 2015*). We identified 178 cortex-like cells based on strong expression of canonical NSPC and neuron marker genes (i.e., NSPCs and neurons: *FOXG1, NFIA, NFIB*; NSPCs: *PAX6, SOX2, GLI3*; neurons: *NEUROD6*) and the lack of expression of the ventral and medial telencephalic markers *OTX2* and *RSPO2* (*Figure 1—figure supplement 2*).

We sub-classified the 178 cerebral cortex-like cells based on the correlation between their transcriptomes and the bulk transcriptomes of laser-capture microdissected VZ, iSVZ, oSVZ, and cortical plate of fetal human neocortex (GSE38805, [*Fietz et al., 2012*]). We found that groups of cells correlated best with one of the four zones, suggesting that the range of cell types present in the human fetal and organoid cerebral cortex are represented in our chimpanzee data (*Figure 1E*). Consistent with this, each chimpanzee cell represents a cell state on a continuum from NPSCs to neurons based on gene expression signatures extracted from fetal human cerebral cortex transcriptomes (*Figure 1F, Figure 1—figure supplement 3*) (*Camp et al., 2015*).

We next classified the chimpanzee cerebral cortex cells by determining the fetal cell type with which each cell most strongly correlates, resulting in 73 APs, 25 BPs, and 80 neurons. Analysis of known cell type markers revealed expression patterns consistent with what has been observed in human organoid and fetal cerebral cortex (*Figure 1G*) (*Camp et al., 2015*). Though this classification is convenient to describe the cell types present in the chimpanzee organoid, we note that many of the cells can be described as intermediates between APs, BPs, and different stages of neuron maturation. We inferred lineage relationships among the chimpanzee cerebral cortex in an adjacency network based on pairwise correlations between cells (*Figure 1H*), revealing a structured topology where VZ-APs connect to cortical plate neurons through SVZ-BPs. These lineage relationships were corroborated using a minimal spanning tree algorithm (*Figure 1—figure supplement 3G*) (*Trapnell et al., 2014*). Together, these data allowed reconstruction of the chimpanzee organoid cerebral cortex from single-cell transcriptomes.

## Chimpanzee and human gene expression in the developing cerebral cortex

To further explore transcriptome similarities and differences between chimpanzee and human cerebral cortex cells, we compared them to the single-cell transcriptomes of 220 fetal human cortex cells (12–13 weeks post-conception (wpc), published in (*Camp et al., 2015*), GSE75140) and 207 cortex-like cells from human cerebral organoids (40–80 days, 155 single-cell transcriptomes published in (*Camp et al., 2015*), GSE75140; 52 single-cell transcriptomes acquired as part of this study) (*Figure 3—source data 1*). In a PCA, the first principal component (PC1) separated NSPCs and neurons, whereas PC2 separated species (*Figure 3A*). Hierarchical clustering of organoid and fetal cells showed that human and chimpanzee organoid and human fetal cells were distributed together within the two main sub-clusters representing NSPCs and neurons (not shown), and showed highly correlated expression of marker gene patterns (*Figure 3B*).

We constructed an intercellular correlation network, which revealed a VZ sub-network of human and chimpanzees APs that link through BPs expressing iSVZ and oSVZ markers to cortical plate neurons. Generally, APs, BPs, and neurons from human and chimpanzee intermixed, confirming that cells in the chimpanzee organoid cortices have a zonal organization consistent with what is observed histologically (*Figure 3C,D*). In conclusion, the major proportion of the variation in these data is not between in vitro and in vivo tissues or between species, but among cell states during neurogenesis, confirming that the major features of the genetic programs regulating the NSPC-to-neuron lineage are conserved between human and chimpanzees, and are recapitulated in cerebral organoids.

To identify genes differentially expressed between chimpanzee and human cortex-like cells, we remapped all single-cell transcriptome reads to a consensus human-chimpanzee genome and used human annotations to identify 1-to-1 orthologous genes. We then used a Bayesian approach to identify differentially expressed genes by comparing cerebral organoid APs and neurons between species (ignoring BPs due to the low number of BPs identified). We identified 297 and 279 genes that were more highly expressed in human APs and neurons, respectively, and 283 and 314 genes that were more highly expressed in chimpanzee APs and neurons, respectively (*Figure 3E, Figure 3—source data 2*). In addition to the between-species comparisons, we identified genes differentially

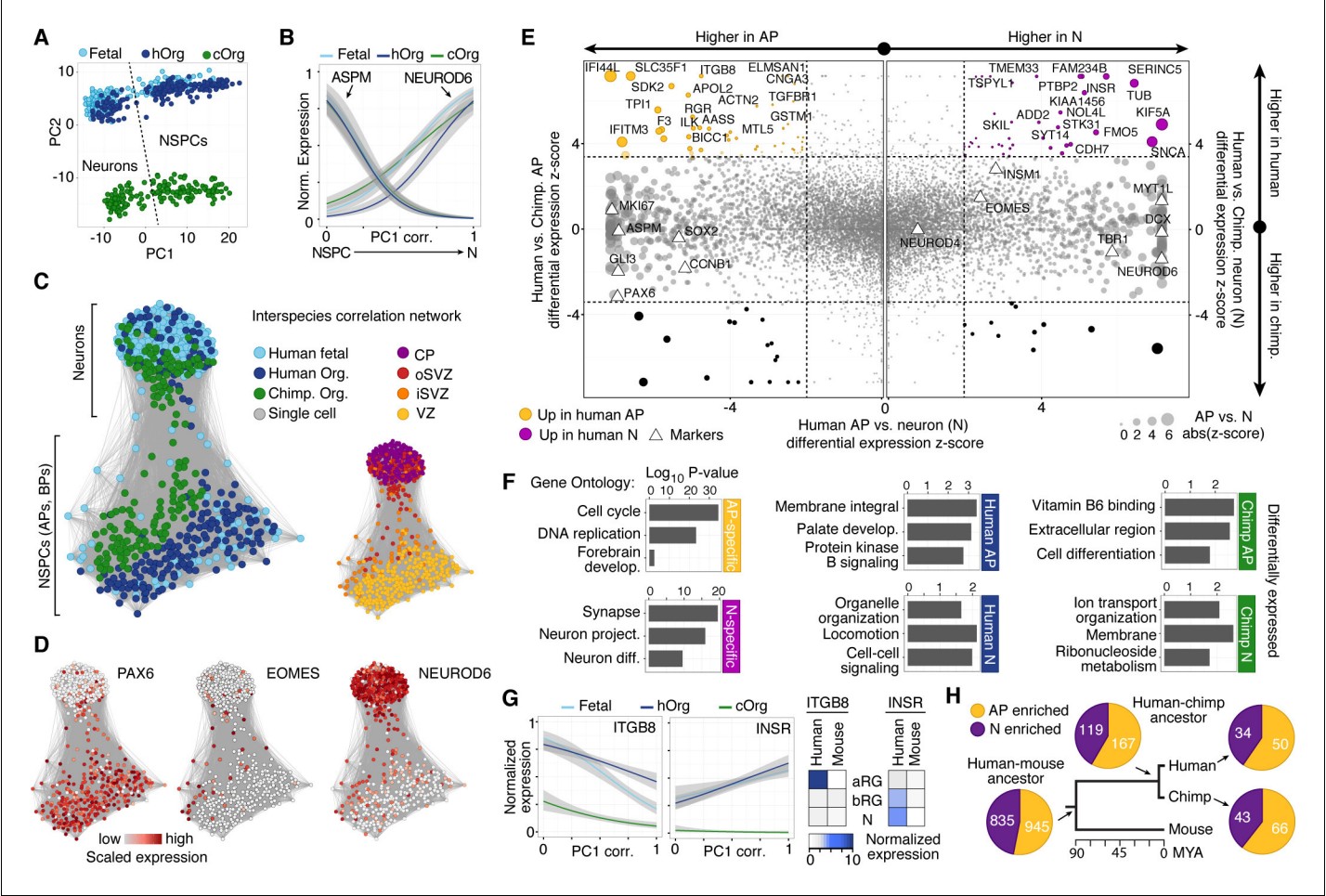

**Figure 3.** Comparing human and chimpanzee cerebral cortex gene expression. (**A**) PC1 and PC2 from PCA separated NSPCs and neurons, and human and chimpanzee, respectively. PCA was performed on all single-cell transcriptomes using genes expressed in more than two cells and with a non-zero variance. (**B**) Quasibinomial fit line of representative marker gene expression across cells ordered by correlation with PC1. (**C**) Lineage network based on pairwise correlations between human fetal, human organoid, and chimpanzee organoid cells reveals a differentiation topology from VZ APs through BPs in iSVZ and oSVZ, to cortical plate (CP) neurons, with inter-species mixing in all stages. (**D**) Lineage network (see (**C**)) coloured by scaled expression level of marker genes. (**E**) Scatterplots showing z-scored significance estimates from single-cell differential expression (SCDE) analysis based on Bayesian probabilistic models. Reads from human and chimpanzee were mapped to a consensus genome, and human gene annotations were used for expression counting. The x-axis represents SCDE between human organoid APs vs. human organoid neurons. The y-axes on the left and right plots represent SCDE between human and chimpanzee APs and neurons (N), respectively. Genes coloured as white triangles represent marker genes from *Figure 1* and are generally not differentially expressed between human and chimpanzee, but do vary between APs and neurons, validating the SCDE analysis. Yellow and purple circles represent genes upregulated specifically in human APs and neurons, respectively. Circles are sized based on differential expression between human APs and neurons. *Figure 3—figure supplement 1* shows a similar plot from the chimpanzee perspective. (**F**) Gene ontology enrichments (-log₁₀ P-value) for differentially expressed gene groups shown in panel E. Left, human APs (yellow) and neurons (N, purple) that are not differential between human and chimpanzee. Center, upregulated in human APs (top) or neurons (N, bottom) compared to chimpanzee. Right, upregulated in chimpanzee APs (top) or neurons (N, bottom) from *Figure 3—figure supplement 1*. (**G**) Left, expression profiles of ITGB8 and INSR are shown from human organoid, chimpanzee organoid, and human fetal cells ordered by correlation with PC1. Right, bulk RNA-seq data from sorted aRG, bRG, and neurons (N) from human and mouse developing neocortex (*Florio et al., 2015*) confirms enriched expression of ITGB8 and INSR in human APs and neurons, respectively. (**H**) The same bulk RNA-seq data was used to confirm and estimate the origin of differential gene expression in APs *versus* neurons from single-cell organoid data. Pie chart shows the proportion of AP-enriched (yellow) or neuron-enriched (N, purple) genes that are observed in human, chimpanzee, and mouse. Pie charts also show the proportion of genes differential between APs and neurons that are observed only in human and chimpanzee, but not mouse (human-chimp ancestor), or genes specific to human or chimpanzee.

The following source data and figure supplement are available for figure 3:

**Source data 1.** Processed single-cell RNA-seq data for human cells.

**Source data 2.** Results of differential gene expression analyses.

*Figure 3 continued on next page*

*Figure 3 continued*

**Figure supplement 1.** Differential expression analysis between chimpanzee and human cerebral cortex cells from the chimpanzee perspective.

expressed between human or chimpanzee APs and neurons to identify cell-type specific genes (for human: 1328 AP-specific, 1132 neuron-specific; for chimpanzee: 1501 AP-specific, 1166 neuron-specific). The vast majority (94%) of genes that are AP-specific and neuron-specific in humans are not differential between human and chimpanzee (*Figure 3E*, *Figure 3—figure supplement 1*). Of the differentially expressed genes between species, we identified 93 genes that are strongly upregulated in human organoid APs and 72 genes upregulated in human organoid neurons. Gene ontology enrichments suggest that the proteins encoded by some of these genes are integral to cell membranes and involved in intercellular signalling (*Figure 3F*, *Figure 3—source data 2*), for example integrin beta 8 (ITGB8) in APs and insulin receptor (INSR) in human neurons. This upregulation of ITGB8 specific to human APs and INSR specific to human neurons is also observed in comparisons between human and mouse (*Florio et al., 2015*) (*Figure 3G*).

When comparing these results to bulk RNA-seq data from mouse APs and neurons (*Florio et al., 2015*), we find that 75% of the genes with expression specific to APs or neurons in humans are also specific to each cell type in the mouse, suggesting that these gene expression programs were already established and likely present in the common ancestor of mouse, human and chimpanzee some 90 million years ago (*Figure 3F*). Notably, a similar proportion of AP- and neuron-specific genes were gained on the chimpanzee and human branch subsequent to their separation, suggesting that our analysis did not have a strong human bias. About 12% of these genes specific to AP or neurons in both human and chimpanzee were not specific to these cell types in the mouse (*Florio et al., 2015*), suggesting that they may be involved in developmental processes specific to the primate cerebral cortex.

## Live imaging of NSPC mitoses in human fetal neocortex and cerebral organoids

We used an established live imaging method (*Mora-Bermudez et al., 2014*) to compare dividing cortical APs, i.e. cells undergoing mitosis at the ventricular surface (presumably mostly aRG), in slice cultures of both 11–13 wpc human fetal neocortex and human D30 cerebral organoids. We did not observe signs of strong perturbation during live image acquisition in either system, such as mitotic arrest (*Figure 4A,C,E*; see also *Figure 5A–C* and *Video 1*) or lack of nuclear movements and cell death. Chromosome dynamics and spindle orientation of APs, as revealed by the orientation of the metaphase plate, were similar in human developing neocortex and human organoids, both before anaphase (*Figure 4A–D,G*) and during anaphase (*Figure 4A–D,H,I*), when cell cleavage initiates. This strongly suggests that cerebral organoids are a suitable model to study live NSPC division and spindle orientation dynamics.

## Spindle orientation dynamics are similar in human and chimpanzee NSPCs

Spindle orientation can determine symmetric vs. asymmetric NSPC division (*Lancaster and Knoblich, 2012*; *Mora-Bermudez and Huttner, 2015*; *Mora-Bermudez et al., 2014*; *Shitamukai and Matsuzaki, 2012*) and is therefore a major candidate mechanism to explain the approximately 3-fold expansion of the neocortex in humans compared to great apes. We compared spindle orientation dynamics between human and chimpanzee APs in cerebral organoids. However, our data revealed no clear differences in spindle orientation, either during metaphase (*Figure 4C–G*) or shortly after anaphase onset (*Figure 4C–F, I–J*). As deduced from the orientation of the chromosome plates, most APs in both human and chimpanzee divided with a cleavage orientation largely perpendicular to the apical, ventricular surface, showing deviations of fewer than 30° from a perfect orthogonal cleavage. Oblique and near-horizontal orientations were also observed, but at a much lower abundance and at similar frequencies in chimpanzee and human organoids (*Figure 4H–J*). This shows

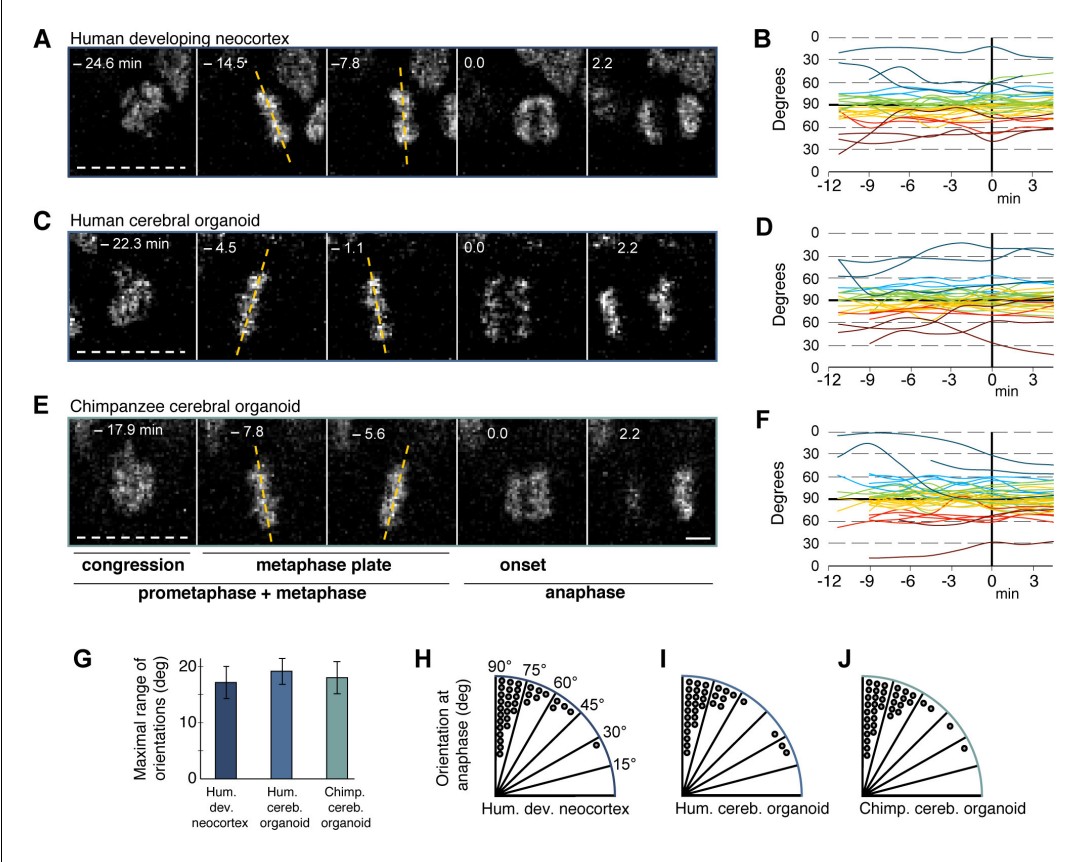

**Figure 4.** Spindle orientation variability is similar between APs of human developing neocortex, human organoids and chimpanzee organoids. Live tissue imaging of spindle orientation, as reported by chromosome plate orientation, in organotypic slice culture of developing neocortex and cerebral organoids. Measurements were started after all chromosomes had formed a tight metaphase plate. 0 min is anaphase onset. Time-lapse is ~1.1 min. (A,C,E) APs in a coronal slice of 13 wpc human frontal neocortex (A), in a slice of a D30 human cerebral organoid from iPSC line SC102A-1 (C), and in a slice of a D30 chimpanzee cerebral organoid from iPSC line Sandra A (E). The time indicated on each image is when that image was taken, relative to anaphase onset (0 min). White dashed lines, ventricular surface. Yellow dashed lines indicate the two metaphase plate orientations with the greatest difference to each other. Scale bar, 5 µm. (B,D,F) Quantification of all orientations of the chromosome plates from the beginning of the metaphase plate stage to anaphase, for APs in the three respective tissues described in (A,C,E). To facilitate tracing, individual tracks are colour-coded according to the initial range of the track, and the 90°−0° range is depicted twice (green and yellow, 90°−75°; cyan and red, 75°−60°; blue and dark red, 60°−0°; 90° indicates perfectly vertical chromosome plates). (G) Maximal range of chromosome plate orientations for APs, from the beginning of the metaphase plate stage to anaphase onset, as determined in the measurements shown in (B,D,F). Data are the mean ± SEM of ≥34 APs from 3 independent experiments each. (H,I,J) Orientation of chromosome plates at 2.2 min after anaphase onset, which indicates the predicted plane of cleavage, as determined in the measurements shown in (B,D,F). 90° indicates a perfectly vertical cleavage plane.

that the frequency of asymmetric cell division caused by oblique spindle orientation is most likely not a major difference between human and chimpanzee APs.

## Longer prometaphase-metaphase in human than great ape APs

We noticed, however, unexpected differences between human and chimpanzee APs in their progression through mitosis. Specifically, measurement of the length of the various phases of mitosis (for details, see Materials and methods) revealed that APs in 11–13 wpc fetal human neocortex and D30 cerebral organoids remained approximately 5 min longer in prometaphase-metaphase than APs in chimpanzee organoids (*Figure 5A–C,E*; *Video 1*). By comparison, prometaphase-metaphase of APs in slice culture of mouse neocortex, a well-characterized model system for neurogenesis, lasted for only approximately half the amount of time than human APs (*Figure 5D,E*; *Figure 5—source data 1*).

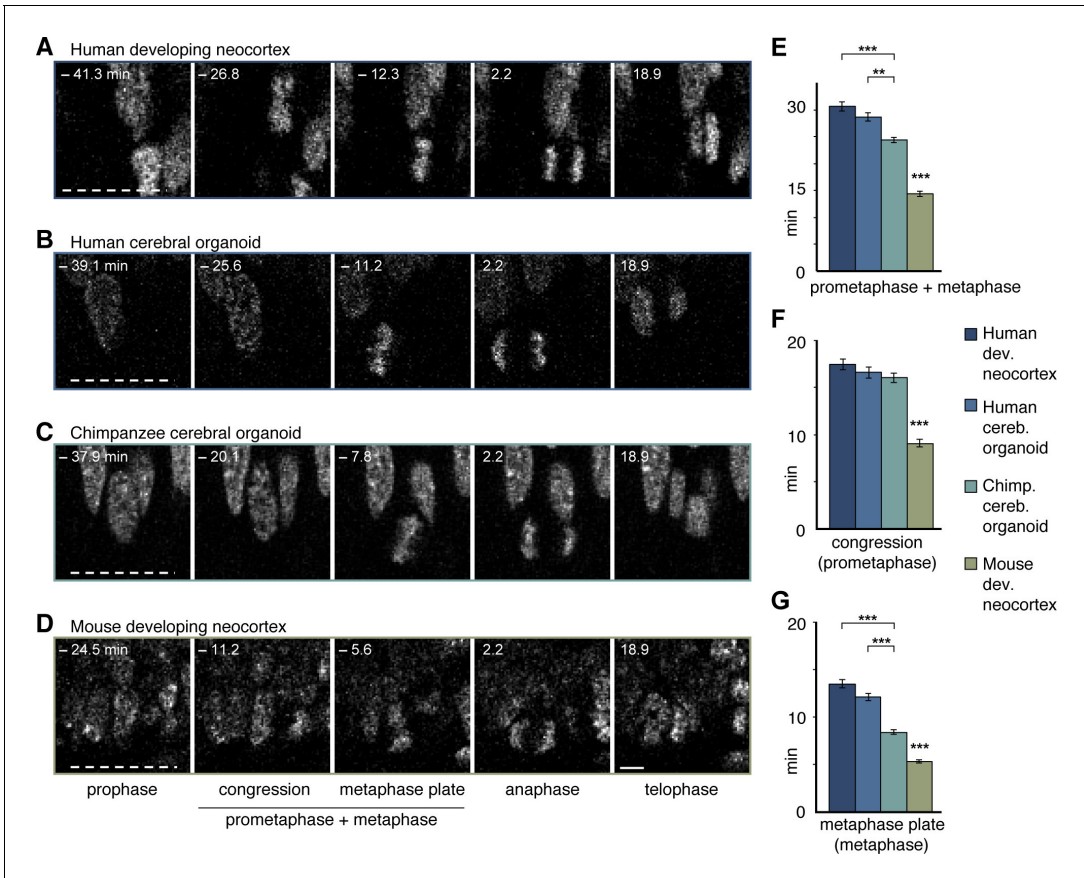

**Figure 5.** Differences in prometaphase-metaphase length between APs of human developing neocortex, human organoids, chimpanzee organoids and mouse developing neocortex. Live tissue imaging of mitotic phases, as reported by chromosomes, in organotypic slice culture of developing neocortex and cerebral organoids. 0 min is anaphase onset. Time-lapse is ~1.1 min. (A–D) APs in a coronal slice of 13 wpc human frontal neocortex (A), in a slice of a D30 human cerebral organoid from iPSC line SC102A-1 (B), in a slice of a D30 chimpanzee cerebral organoid from iPSC line Sandra A (C), and in a coronal slice of E14.5 mouse neocortex. The time indicated on each image is when that image was taken, relative to anaphase onset (0 min). White dashed lines, ventricular surface. Scale bar, 5 μm. (E–G) Time between the start of chromosome congression and anaphase onset (referred to as 'prometaphase + metaphase') (E), between the start of chromosome congression and the formation of a metaphase plate (referred to as 'prometaphase') (F), and between the formation of a metaphase plate and anaphase onset (referred to as 'metaphase') (G), for APs in the four tissues described in (A–D). Data include APs from 11–13 wpc human neocortex, organoids from the human iPSC lines SC102A-1 and 409b2, and chimpanzee iPSC lines Sandra A and PR818-5, and are the mean ± SEM of ≥60 APs from ≥4 independent experiments each. Bracket with **p<0.01; brackets with ***p<0.001; ***p<0.001 (mouse vs. all primate tissues).

The following source data and figure supplements are available for figure 5:

**Source data 1.** Durations of all mitotic phases.

**Figure supplement 1.** The length of the mitotic phases other than prometaphase-metaphase is similar between human and chimpanzee APs.

**Figure supplement 2.** Differences in prometaphase-metaphase length between APs of D30 and D52 human and chimpanzee cerebral organoids.

**Figure supplement 3.** Prometaphase-metaphase in orangutan organoid APs is similar to chimpanzee organoid APs.

**Figure supplement 4.** Determination of cell cycle parameters of human and chimpanzee organoid APs using cumulative EdU labeling.

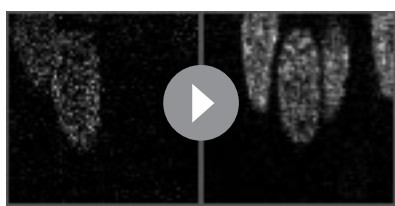

**Video 1.** Differences in prometaphase-metaphase length between APs of human and chimpanzee cerebral organoids. Related to *Figure 5B and C* Live tissue imaging of mitotic phases, as reported by chromosomes, in organotypic slice culture of cerebral organoids. Time-lapse is ~1.1 min. Datasets are the same as in *Figure 5B and C*. Left side: APs in a slice of a D30 human cerebral organoid from iPSC line SC102A-1. Right side: APs in a slice of a D30 chimpanzee cerebral organoid from iPSC line Sandra A. Growing colour bars at the bottom indicate time progression of the respective dividing AP and are synchronized to the beginning of prometaphase (in green). Metaphase plate time is in yellow and anaphase time is in red. Note the slower progression of the dividing human AP on the left.

To trace the specific phase of mitosis when this difference arises, we used chromosome morphology and dynamics to determine the time chromosomes spent congressing toward the equatorial plane of the cell (defined here as 'prometaphase') and the time they spent tightly aligned as a metaphase plate (defined here as 'metaphase'). Remarkably, the longer prometaphase-metaphase of human than chimpanzee APs was specifically due to a ~40-60% lengthening of metaphase (*Figure 5A–C,G*), whereas prometaphase was not significantly different (*Figure 5A–C,F*; *Video 1*). By contrast, in mouse APs, both prometaphase and metaphase were found to be significantly shorter than the respective mitotic phases in human and chimpanzee APs (*Figure 5D,F,G*; *Figure 5—source data 1*).

None of the other mitotic phases (prophase, anaphase, telophase) differed in length between APs in human fetal neocortex and human cerebral organoids vs. chimpanzee organoids. However, anaphase of mouse APs was found to be significantly shorter than that of human and chimpanzee APs (*Figure 5—figure supplement 1A*; *Figure 5—source data 1*). These differences between species in the individual mitotic phases were reflected in the cumulative length of total mitosis, which was significantly shorter in mouse APs than human and chimpanzee APs (*Figure 5—figure supplement 1B*).

To search for potential functional implications of these observations, we next quantified and compared the length of prometaphase-metaphase in human and chimpanzee APs of day 52 (D52) cerebral organoids, and compared the results with those of D30 organoids. Prometaphase-metaphase (*Figure 5—figure supplement 2A*) and metaphase alone (*Figure 5—figure supplement 2C*; *Figure 5—source data 1*) were shorter in D52 than in D30 human APs, and not anymore statistically significantly different in length from D52 chimpanzee APs. The longer metaphase of human than chimpanzee organoid APs may therefore characterise early phases of cortical development, when proliferative AP divisions are predominant.

We also generated cerebral organoids from an orangutan iPSC line and determined the length of AP prometaphase-metaphase. This revealed that the length of prometaphase-metaphase in orangutan D30 organoid APs was similar to that of chimpanzee APs and significantly shorter than that of human organoid APs (*Figure 5—figure supplement 3A,B*). As was the case for the human-chimpanzee AP comparison, the shorter prometaphase-metaphase of orangutan than human APs was due to a shorter metaphase (*Figure 5—figure supplement 3A,D*) rather than prometaphase (*Figure 5—figure supplement 3A,C*; *Figure 5—source data 1*). Together, these data indicate that human APs specifically lengthen prometaphase-metaphase as compared to great ape APs.

In light of these differences in the duration of mitotic phases, it was of interest to compare the length of the total cell cycle of human and chimpanzee organoid APs. Using cumulative EdU labelling of D52-D54 cerebral organoids (*Figure 5—figure supplement 4A*), we found a relatively minor (~6%) difference in total cell cycle length, with human APs (PAX6+TBR2– cells) exhibiting a ~2.7 hr longer cell cycle (46.5 h) than chimpanzee APs (43.8 h) (*Figure 5—figure supplement 4B*). However, a notable difference between the two species was the length of S-phase, which was nearly 5 hr longer in human (17.5 h) than chimpanzee (12.8 h) organoid APs (*Figure 5—figure supplement 4B*).

## The prometaphase-metaphase lengthening in humans occurs upon neural differentiation

To investigate the origin of the longer metaphase in human than chimpanzee APs, we measured mitotic phase lengths in the original iPSCs used to grow the cerebral organoids. This revealed that

both the human and chimpanzee organoid APs had a longer prometaphase-metaphase than their respective iPSCs of origin, showing that this general lengthening was due to the transition between iPSCs and the organoids of both species (*Figure 6A,B,E*). In human APs, however, the lengthening was greater than in chimpanzee APs. In contrast to APs, human and chimpanzee iPSCs had similar prometaphase-metaphase lengths (*Figure 6A,B,E*; *Figure 5—source data 1*).

Further dissection into individual phases revealed that, whereas both human and chimpanzee APs had a longer prometaphase than their iPSCs of origin (*Figure 6A,B,F*), only human APs had a longer metaphase when compared to the iPSCs of origin (*Figure 6A,B,G*; *Figure 5—source data 1*).This shows that prometaphase-metaphase lengthened in both species as APs were generated during cerebral organoid formation with the accompanying neural differentiation. However, the lengthening characteristics were species-specific. The lengthening was greater in humans than chimpanzees because the metaphase plate stage became longer only in human APs.

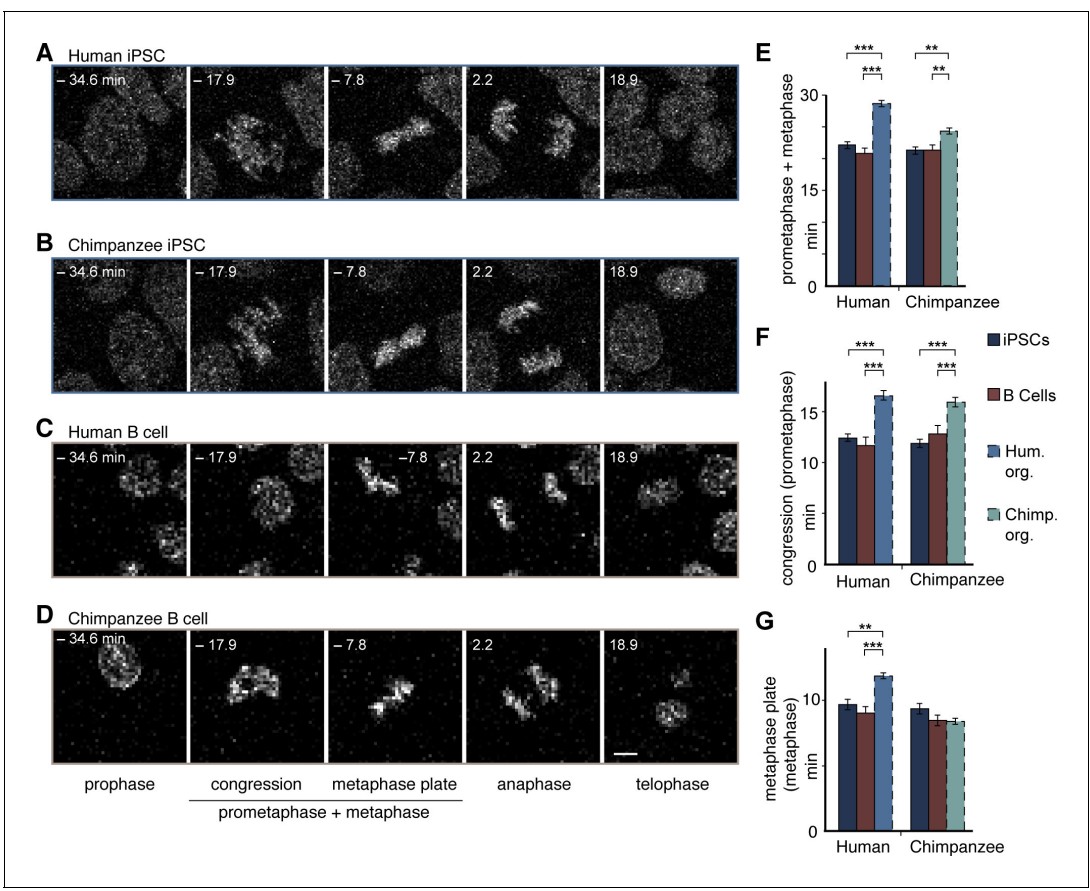

**Figure 6.** Human and chimpanzee organoid APs exhibit longer prometaphase, and human organoid APs longer metaphase, than their iPSC lines of origin or B cells. Live imaging of mitotic phases, as reported by chromosomes, in human and chimpanzee iPSCs and B cells. 0 min is anaphase onset. Time-lapse is ~1.1 min. (A–D) Human iPSC (SC102A-1) (A), chimpanzee iPSC (Sandra A) (B), human B cell (A158) (C), and chimpanzee B cell (Dorien) (D). The time indicated on each image is when that image was taken, relative to anaphase onset (0 min). Scale bar, 5 μm. (E–G) Time between the start of chromosome congression and anaphase onset (referred to as 'prometaphase + metaphase') (E), between the start of chromosome congression and the formation of a metaphase plate (referred to as 'prometaphase') (F), and between the formation of a metaphase plate and anaphase onset (referred to as 'metaphase') (G). Data include cells from each of the following iPSC lines: human, SC102A-1 and 409b2; chimpanzee, Sandra A and PR818-5; and from the following B cell lines: human, A144, A156 and A158; chimpanzee, Jahaga, Ulla and Dorien. For comparison, the relevant mitotic phase lengths of human and chimpanzee cerebral organoid APs from *Figure 5* are shown (columns with dashed line). Data are the mean ± SEM of ≥30 cells from ≥3 independent experiments each. **p<0.01; ***p<0.001.

The following figure supplement is available for figure 6:

**Figure supplement 1.** The length of prophase, anaphase and telophase is similar in human and chimpanzee iPSCs, B cells and organoid APs.

To determine whether prometaphase-metaphase length may differ between chimpanzees and humans also in another cell type, we measured mitotic phases in human and chimpanzee B cells. In contrast to fetal tissue, these cells can be obtained not only from humans but also chimpanzees by collecting blood, that is, without major invasive procedures. The length of prometaphase-metaphase in B cells, as well as prometaphase and metaphase measured individually, were similar to that in iPSCs (*Figure 6 C–G*), and hence significantly shorter than in human or chimpanzee APs. By contrast, the other mitotic phases were similar among organoid APs, iPSCs and B cells in both species (*Figure 6—figure supplement 1*; *Figure 5—source data 1*). This raises the intriguing possibility that lengthening of prometaphase-metaphase could be specific to ape and human NSPCs and, furthermore, that lengthening of the metaphase plate time could be specific to human NSPCs.

## Longer prometaphase-metaphase in proliferative than neurogenic mouse APs

To investigate potential functions of prometaphase-metaphase lengthening, we asked whether mitotic phases were different between proliferating and neurogenic APs. To this end, we measured mitotic phase lengths in a transgenic mouse line where EGFP is expressed under the promoter of the pan-neurogenic marker Tis21 in neurogenic but not proliferative NSPCs (*Haubensak et al., 2004*; *Iacopetti et al., 1999*). This revealed that prometaphase-metaphase was longer in proliferative AP divisions (Tis21– ) than in neurogenic AP divisions (Tis21+ ), whereas the separate phases were not significantly different (*Figure 7*; *Figure 5—source data 1*). These results suggest that a lengthening of prometaphase-metaphase may be characteristic of proliferating NSPCs.

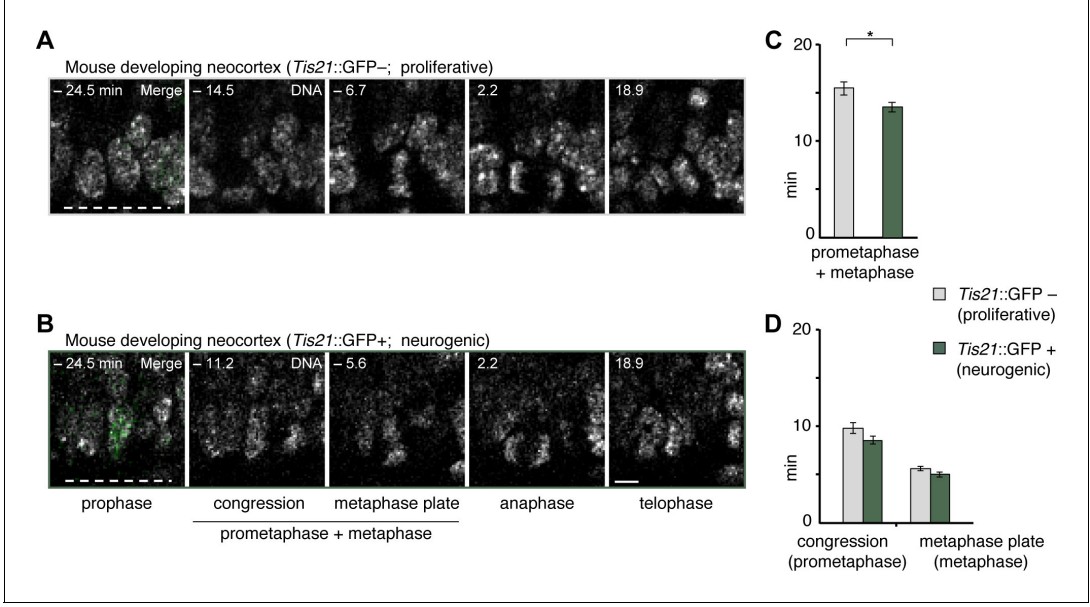

**Figure 7.** Prometaphase-metaphase is longer in proliferative than neurogenic mouse APs. Live tissue imaging of mitotic phases, as reported by chromosomes, in organotypic slice culture of E14.5 mouse neocortex. 0 min is anaphase onset. Time-lapse is ~1.1 min. Data is from the same mouse dataset shown in *Figure 5*, but distinguishes between *Tis21*::GFP– (proliferative) and *Tis21*::GFP (neurogenic) APs. (A,B) APs in a coronal slice of mouse E14.5 dorsolateral telencephalon, either negative (A) or positive (B) for expression of *Tis21*::GFP. The time indicated on each image is when that image was taken, relative to anaphase onset (0 min). White dashed lines, ventricular surface. Scale bar, 5 μm. Image panels in (B) are the same as in *Figure 5D*, but the *Tis21*::GFP fluorescence (green) is included in the prophase image (merge). The GFP channel is also merged in the prophase image of (A), and the other panels are DNA staining only. (C,D) Length of prometaphase and/or metaphase in proliferative vs. neurogenic APs. Data are the mean ± SEM of 41 *Tis21*::GFP– and 37 *Tis21*::GFP APs from 4 independent experiments. *p<0.05. (C) Time between the start of chromosome congression and anaphase onset (referred to as 'prometaphase + metaphase'). (D) Time between the start of chromosome congression and the formation of a metaphase plate (referred to as 'prometaphase', left), and time between the formation of a metaphase plate and anaphase onset (referred to as 'metaphase', right).

## Gene expression in human and chimpanzee mitotic APs

We used the single-cell RNA-seq data to identify organoid APs in different phases of the cell cycle (*Figure 8A*, *Figure 8—figure supplement 1*) and searched for genes that might be involved in human-specific lengthening of the metaphase. We compared human organoid APs in G1 with APs in G2-M and identified 395 genes with enriched expression in G2-M (*Figure 8B*). We next compared human APs in G2-M with human iPSCs (TkDA3-4) and an endothelial cell line (HUVEC; both single-cell RNA-seq data sets in GSE81252) to understand the specificity of G2-M regulation in APs. We found that nearly all genes upregulated in human APs in G2-M compared with human APs in G1 were also upregulated during G2-M in iPSCs and endothelial cells (*Figure 8C*). Therefore, the expression level of these genes is unlikely to contribute to the specificity of mitotic control of human APs in G2-M. However, we identified many genes that were highly expressed throughout the human AP cell cycle and were specific to APs. Genes with the highest specificity score encoded canonical cerebral cortex patterning transcription factors such as PAX6, ID4, and GLI3, as well as proteins involved in cell adhesion and ECM signalling (CDH4, EFNB1/2, COL4A2). Notably, no genes associated with cell cycle, kinetochore, or spindle terms were specific to human APs (*Figure 8C*, inset). Of genes specific to APs, a subset were also differentially expressed between human and chimpanzee cerebral organoids (APOLD1, BICC1, EFNB1, GSTM1, IFI44L, ITGB8, SDK2, SEMA5A, SLC35F1, ZNF516), which makes them candidates for the unique regulation of AP proliferation in humans (*Figure 8D*).

## Discussion

We have characterized cerebral organoids generated from chimpanzee iPSCs, including a newly generated iPSC line, and shown that their cytoarchitecture, cell type composition, and neurogenic gene expression programs are remarkably similar to human cerebral organoids and to human fetal neocortex. This extends a very recent study (*Otani et al., 2016*) and establishes cerebral organoids as a valid system to compare human and chimpanzee NSPC behaviour. Using this system, we have shown that human and chimpanzee APs differ in that prometaphase-metaphase is longer in humans than in chimpanzees. This difference was also observed between human and orangutan and reflects a greater extent of prometaphase-metaphase lengthening that occurs as human APs are generated during cerebral organoid development from IPSCs. There are two intriguing implications as to the biological significance of this prometaphase-metaphase lengthening in human APs.

One is related to the fate of the progeny arising from AP division. Mouse *Tis21*::GFP-negative APs, which are known to undergo proliferative divisions to generate more APs, have a longer prometaphase-metaphase than *Tis21*::GFP-positive APs, which are known to undergo neurogenic divisions to generate BPs (*Haubensak et al., 2004*). The longer prometaphase-metaphase in human than chimpanzee APs would therefore be consistent with a greater tendency for proliferative than neurogenic divisions. In this respect, other changes in progeny fate have also been recently observed in a different context, upon an experimentally induced and considerable prolongation of AP mitosis in embryonic mouse neocortex (*Pilaz et al., 2016*).

Another set of observations are consistent with the notion that the longer prometaphase-metaphase in human than chimpanzee APs may indicate a greater tendency for proliferative than differentiative divisions. The human vs. chimpanzee prometaphase-metaphase difference decreased in the course of organoid cortical development from D30 to D52, when one would expect proliferative AP divisions to decrease and differentiative AP divisions to increase.

Further support for this notion was obtained by analysis of the interphase of the cell cycle, specifically S-phase. Mouse *Tis21*::GFP-negative (proliferative) APs have previously been shown to have a longer S-phase than *Tis21*::GFP-positive (differentiative) APs (*Arai et al., 2011*). The substantially longer S-phase of human than chimpanzee APs observed here is therefore also in line with human APs having a greater tendency for proliferative divisions.

Finally, the changes in the abundance of NSPC types in the course of cerebral organoid development yielded data supporting a greater AP proliferation in human than chimpanzee. Specifically, the proportion of PAX6+TBR2– NSPCs, located in the VZ and thus constituting proliferating APs, decreased in both human and chimpanzee cerebral organoids, but the value reached in human organoids was slightly higher than that in chimpanzee organoids (*Figure 2B*). Conversely, the proportion of PAX6+TBR2+ NSPCs, located in the basal VZ and SVZ and constituting BPs with neurogenic

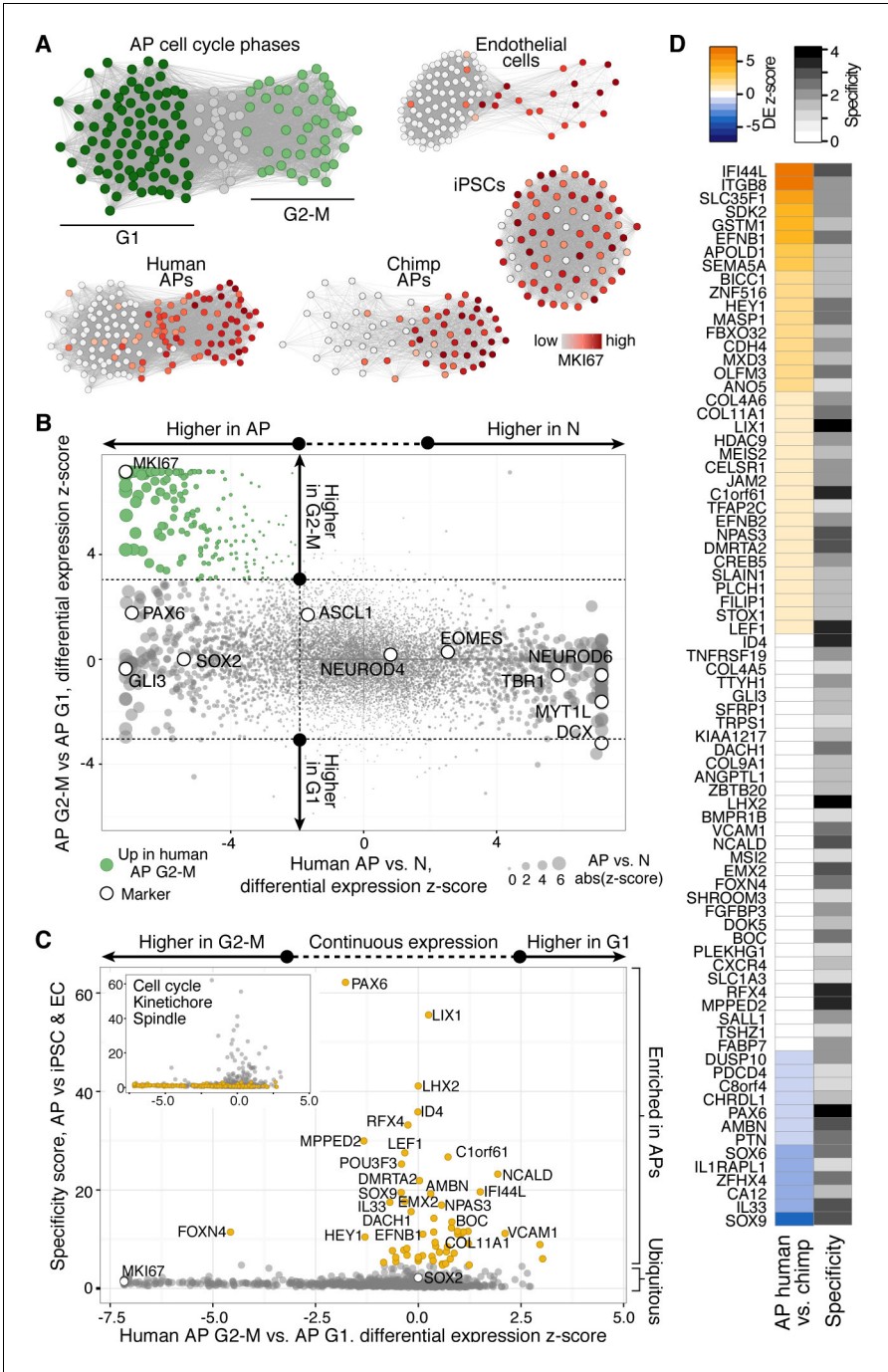

**Figure 8.** Differential gene expression during AP mitotic phases. (**A**) G1 (dark green) and G2-M (light green) cell cycle phases were assigned to cells by performing PCA using genes involved in cell cycle regulation. PC1 and PC2 described cell cycle phases, and the top 50 correlating and anticorrelating genes were used to infer an intercellular correlation network for human and chimp APs, human iPSCs, and a human endothelial cell line. Networks are coloured based on the expression level of MKI67. (**B**) Scatterplot shows z-scored significance estimates from single-cell differential expression (SCDE) analysis between human organoid APs vs. neurons (N, x-axis) and APs in G2-M vs. APs in G1 (y-axis). Genes coloured as white circles represent marker genes and green circles represent genes upregulated specifically in APs in G2-M. Circles are sized based on differential expression between human APs and neurons. (**C**) iPSC and endothelial cell (EC) expression was used to assign a specificity score for genes enriched in human organoid APs compared to neurons (higher in AP genes from panel B). The specificity score is plotted against the differential expression between APs in G2-M and APs in G1. Cells with high AP specificity scores are in yellow in the main scatter plot. This shows that nearly all genes enriched in G2-M phase

*Figure 8 continued on next page*

*Figure 8 continued*

of the AP cell cycle are not specific to APs, but also enriched in G2-M of mitotic iPSCs and endothelial cells. (D) Heatmap shows the differential expression score between human and chimpanzee APs (z-score) and AP specificity score (Log$_2$ normalized) of the same genes that are specific to APs relative to endothelial cells and iPSCs.

The following figure supplement is available for figure 8:

**Figure supplement 1.** Cell cycle assignment for differential gene expression analysis.

potential, showed a greater increase in chimpanzee than human cerebral organoids. In sum, two independent lines of evidence, the detailed analysis of AP mitosis phase lengths and the determination of the proportions of the various NSPC types, support the concept that a longer neurogenic period (*Lewitus et al., 2014*), which in turn implies a longer phase of NSPC proliferation (*Otani et al., 2016*), contributes to the greater expansion of the neocortex in humans than the great apes.

The second implication as to the biological significance of the longer prometaphase-metaphase in human than chimpanzee APs concerns the fact that these are the phases of mitosis when chromosomes prepare for segregation, to ensure that only one copy of each chromosome is distributed to each nascent daughter cell (*Musacchio and Salmon, 2007*). The longer duration of prometaphase-metaphase in human than chimpanzee APs, in particular of the metaphase plate stage (*Figure 5B*), may therefore reflect some difference between the two species with regard to the preparation for chromosome segregation.

If the longer prometaphase-metaphase in human than chimpanzee APs reflects a greater tendency for proliferative than neurogenic divisions in the human NSPCs, why did we not detect significant differences between human and chimpanzee APs in spindle orientation, a parameter previously shown to affect the mode of AP division (*Lancaster and Knoblich, 2012*; *Mora-Bermudez and Huttner, 2015*; *Mora-Bermudez et al., 2014*; *Shitamukai and Matsuzaki, 2012*)? This may be due to spindle orientation variability between individual APs being greater than potential inter-species differences. This suggests that, in the cell types and stages analysed, spindle orientation may not play a key role in human vs. chimpanzee neurogenesis. Alternatively, this may reflect the fact that differences in proliferative *versus* neurogenic AP divisions can occur without a change in spindle orientation (*Konno et al., 2008*; *Kosodo et al., 2004*; *Mora-Bermudez and Huttner, 2015*). In this context, differences between human and chimpanzee NSPCs of relevance for neocortex expansion are likely to be small. Consistent with this view, our single-cell transcriptome analyses revealed only few differences between human and chimpanzee, and the differences in the proportions of organoid NSPC populations were in the range of a few percentage points. Furthermore, the ~5 min longer prometaphase-metaphase in human than chimpanzee APs constituted only a fraction of the total duration of their mitosis. These small differences nevertheless provide a set of clues as to which NSPC features may underlie the differential extent of neocortex expansion in humans versus apes, and are consistent with a scenario in which the accumulation of such small differences during evolution may have resulted in the distinct chimpanzee and human neocortices.

## Materials and methods

### Neocortex tissue

Human fetal brain tissue (11–13 weeks post conception (wpc)) was obtained with informed written maternal consent followed by elective pregnancy termination, and neocortex was dissected at room temperature, as described previously (*Florio et al., 2015*). Research involving human fetal brain was approved by the Ethical Review Committee of the Universitätsklinikum Carl Gustav Carus of the Technische Universität Dresden (reference number EK100052004). In addition, research was approved by the Institutional Review Board of the Max Planck Institute of Molecular Cell Biology and Genetics.

Mice were kept pathogen-free at the Biomedical Services Facility of the MPI-CBG. All mouse embryos were heterozygotes of the *Tis21*::GFP knock-in line (*Haubensak et al., 2004*). Imaging was

performed in the dorsolateral telencephalon of E14.5 embryos, at a medial position along the rostro-caudal axis. Embryonic day (E) 0.5 was defined as noon of the day of vaginal plug identification. All experiments using mice were performed according to the German Animal Welfare Legislation.

## Cell lines and organoid culturing

Two human iPSC lines (409b2, SC102A-1), two chimpanzee iPSC lines (PR818-5, Sandra A), and one orangutan iPSC line (Toba) were used to generate cerebral organoids in this study. 409b2 was purchased from the RIKEN BRC cell bank and SC102A-1 was purchased from System Biosciences. PR818-5 (0818) was obtained as a kind gift from F. Gage (*Marchetto et al., 2013*) from the Salk Institute for Biological Studies (La Jolla, CA). Sandra A and Toba were generated in collaboration with Shinya Yamanaka following a nonviral transfection method (*Okita et al., 2013*). Briefly, blood was collected from a chimpanzee and an orangutan, both housed at the Leipzig Zoo, and leukocytes were isolated by Ficoll gradient centrifugation, which were then used for reprogramming to iPSCs. DNA sequencing revealed no chromosome aberrations, and RNA-seq and immunohistochemistry confirmed pluripotent gene and protein expression signatures. Primate blood samples used to generate iPSCs were obtained by certified veterinarians during annual medical examinations or other necessary medical interventions, meaning that no invasive procedures were performed on primates for the sole purpose of our research project. The Max Planck Institute for Evolutionary Anthropology has an institutional permit for the transport of biological material derived from endangered species (DE216-08, see http://cites.org/common/reg/si/e-si-beg.shtml). Human iPSC line TkDA3-4 (*Takebe et al., 2013*) was used to generate iPSC single-cell transcriptomes. iPSC lines were cultured under standard iPSC culturing methods on matrigel (BD Biosciences) using mTeSR1 (Stemcell Technologies). Human endothelial cells. (HUVECs, Lonza, Basel, Switzerland) were maintained in endothelial growth medium (EGM) (Lonza) at 37°C in a humidified 5% $CO_2$ incubator. Single cell transcriptome analysis confirmed the identity of human and chimpanzee iPSCs and human endothelial cells, and showed no contamination with other cell lines. B-cell lines were generated from blood obtained from three human (A144, A156, A158) and three chimpanzee (Dorien, Jahaga, Ulla) individuals. Withdrawal and processing of blood samples was performed according to approved protocols, and was performed for chimpanzee during necessary veterinary interventions. Lymphocytes were isolated from blood using a Ficoll gradient centrifugation. Immortalization was performed by adding Epstein Barr virus (EBV) supernatant to the lymphocytes and further cultivation of the cells until colonies of immortalized B-lymphocytes were established (*Tosato and Cohen, 2007*). B-cells were maintained in RPMI with 10% FBS, 1% Glutamax and 2% penicillin/streptomycin. Cell lines were regularly tested for mycoplasma using a PCR-based test (Minerva Biolabs) and found to be negative.

Human and chimpanzee cerebral organoids were generated from the above iPSCs and cultured for the indicated times as described previously for human cerebral organoids (*Lancaster and Knoblich, 2014*; *Lancaster et al., 2013*), with minor modifications (*Camp et al., 2015*).

## Single-cell RNA-seq experiments
### Preparation of single-cell suspensions from cerebral organoids
To generate single-cell suspensions, cerebral organoids were either dissociated as a whole or first sliced using a vibratome to dissect cortical regions. Whole organoids were washed three times in PBS and incubated at 37°C in 2 ml Accutase (Sigma) plus 2 U/µl DNAse I (New England Biolabs) for ~45 min. For dissections, organoids were washed using PBS and embedded into 4% low-melting agarose (Sigma) and sliced into 150-µm sections using a vibrating microtome (Ci 7000 smz, Camden Instruments). Slices were kept in differentiation plus vitamin A (Diff +VA) medium (*Lancaster et al., 2013*) and inspected under a stereomicroscope (Leica) to dissect cortical regions. Selected regions were washed three times with PBS and incubated in ~200 µl Accutase with DNAse I at 37°C for ~45 min. Additional mechanical dissociation was performed by triturating the tissue. Subsequently, cells were filtered through a 30-µm cell filter (Miltenyi Biotec), washed with Diff +VA medium and spun down at 300 × g for 5 min. The resulting pellet was resuspended in 30–50 µl (for cortical slices) or 250–500 µl (for whole organoids) of Diff +VA medium. In case of excessive debris being present, cells were cleaned using a Percoll (Sigma) gradient centrifugation and the resulting pellet was resuspended in 30–50 µl Diff +VA medium. Counting of cells was performed using a Countess automated

cell counter (Invitrogen) and by staining with Trypan blue. For single-cell experiments, cell suspensions were diluted to a final concentration of 450–600 cells/µl.

## Single-cell cDNA generation

These steps were performed as described (*Camp et al., 2015*; *Treutlein et al., 2014*). Depending on the size distribution of the cells, cells were loaded at a concentration of 250–500 cells per µl onto small (5–10 µm) or medium (10–17 µm) integrated fluidic circuits (IFCs, Fluidigm). Lysis, reverse transcription and amplification were performed on the Fluidigm C1 platform using the SMARTer Ultra Low RNA Kit for the Fluidigm C1 system. External RNA Control Consortium (ERCC) spike-ins (Ambion) were added to the lysis mix at a dilution of 1:80,000. Resulting cDNA was quantified and checked for its size distribution using a capillary gel electrophoresis system (Fragment Analyzer, Advanced Analytical, 1–6000 bps High Sensitivity).

## RNA-Seq library preparation and sequencing

Each cell's cDNA was diluted and libraries were prepared using Nextera XT DNA library preparation kits (Illumina). Up to 96 single-cell libraries were pooled and cleaned up using solid phase reversible immobilization (SPRI) beads (Thermo Scientific). Quantification and library size distribution was assessed on a Bioanalyzer (Agilent) platform using High Sensitivity DNA chips. Up to 192 cells were pooled and sequenced in 100-bp paired-end mode on one lane of an Illumina HiSeq 2500 platform (rapid mode).

## Read processing, mapping and gene quantification

Base-calling, adaptor trimming and demultiplexing of reads was performed using a custom pipeline based on freeIbis (*Renaud et al., 2013*), leeHom (*Renaud et al., 2014*) and deML (*Renaud et al., 2015*). Demultiplexed reads were mapped using TopHat v2.0.14, and FPKM (Fragments Per Kilobase of transcript per Million mapped reads) values per gene were quantified using Cufflinks v.2.2.1 (*Trapnell et al., 2012*). Human reads were mapped to the hg38 reference genome (release 77) and chimpanzee reads were mapped to panTro4 (release 80). The raw FPKM data of all single cells were combined into one master table and transformed to log2 (FPKM +1). (*RStudio Team, 2015*) was used to run (*R Development Core Team, 2010*), scripts to perform principal component analysis (PCA, FactoMineR package), hierarchical clustering (stats package), differential expression analysis (SCDE package), and to construct heatmaps, scatter and line plots, dendrograms, bar graphs, pie charts and histograms. Generally, ggplot2 and gplots packages were used to visualize the data. Gene ontology enrichment analyses were performed using DAVID informatics Resources 6.7 of the National Institute of Allergy and Infectious Diseases (*Huang et al., 2009*).

## Analysis of chimpanzee single-cell RNA-seq data

The Seurat package (*Macosko et al., 2015*) implemented in R was used to identify cell populations present in chimpanzee organoids (*Figure 1—figure supplement 2*). T-distributed stochastic neighbour embedding (tSNE) was performed on all chimpanzee organoid cells using the most significant genes (p-value <10–3, with a maximum of 200 genes per principal component) that define the first 6 principal components of a PCA analysis on the data set. In *Figure 1E* we calculated for each chimpanzee organoid cortex cell the Spearman correlation of its transcriptome (all genes) with bulk transcriptome data from each of 4 microdissected human cortical zones (VZ, iSVZ, oSVZ, CP, mean expression value of each gene across 4 replicates from 13 weeks post conception, data published in ([*Fietz et al., 2012*] GSE38805). We hierarchically clustered (Pearson's correlation distance metric) cells based on their correlation coefficient with germinal zones and visualized the clustering in a heatmap showing correlation coefficients scaled across zones (mean-centering and dividing by standard deviation). The scaling enables a better comparison between cells, since the maximum and minimum correlation for each cell is color-coded in the same way after scaling. We used this analysis to identify the zone with which each individual cell had a maximum correlation.

NSPC and neuron signatures (*Figure 1—figure supplement 3*, *Figure 1F*) were defined by the top 100 genes correlating or anti-correlating with PC1 from PCA of human fetal neocortex, respectively. Each fetal, human organoid, and chimpanzee organoid cortex cell was scored for the NSPC or neuron signature by summing the number of genes from each signature that have an expression

greater than log2 FPKM of 5, and normalizing by the number of all genes expressed above log2 FPKM of 5 for each cell. Lineage network analysis and visualizations were done using igraph implemented in R (http://igraph.sf.net). To construct the chimpanzee cellular network, we computed a pairwise correlation matrix for all chimpanzee cerebral cortex cells and using genes discovered in PCA of fetal neocortex single cell transcriptomes (*Camp et al., 2015*).These same genes had been used to infer lineage relationships in the fetal neocortex. We then generated a weighted adjacency network graph using the graph.adjacency() command and visualized cells as vertices connected to other cells via edges if the pairwise correlation between two cells was higher than 0.4. The fruchterman reingold layout was used to plot the network graph. The combined species network was constructed in a similar way using the same genes and a correlation threshold of 0.4, and was based on FPKM quantification of alignments to each respective species' reference genome. Monocle (*Trapnell et al., 2014*) was used to establish pseudotime estimates and corroborate lineage relationships of chimpanzee cerebral cortex cells using the same genes as in the network analysis.

## Human-chimpanzee consensus genome construction

We re-aligned reads from each cell to a human-chimpanzee consensus genome to account for mapping bias originating from the different genome qualities of the human and chimpanzee genome. The consensus genome was generated as previously described (*He et al., 2014*). In brief, the consensus genome was constructed based on the chained and netted pairwise alignment of human (hg38) and chimpanzee (panTro4) obtained from UCSC. Discordant sites and indels including 6 bp upstream and downsteam of the indel position were masked (replacing the base with N). STAR v2.5.1a (*Dobin et al., 2013*) was used to map all sequences to the consensus genome requiring a minimal fraction of 85% of mapped bases per read. For quantification, HTSeq (*Anders et al., 2015*) v0.6.1.p1 was used applying the human GENCODE v.24 annotation. Resulting count files were combined into one master table containing all cells and genes.

## Differential gene expression analysis

To identify differentially expressed genes between human and chimpanzee, cells were first annotated as AP, BP or neuron based on the fetal cortex cell type with which each cell maximally correlated. After cell type assignment, SCDE (Single Cell Differential Expression) (*Kharchenko et al., 2014*), a Bayesian approach for finding differentially expressed genes accounting for noise inherent to single-cell data, was used to compare the orthologous cell type between human and chimpanzee. AP or neuronal specificity was defined as one standard deviation from the mean of z-scores from SCDE of APs and Neurons (Z.x). A more stringent threshold of twice the standard deviation of the z-score was used to define differential expression between human and chimpanzee (Z.y). For the differential gene expression analysis during mitotic phases, we aimed to identify relatively homogeneous clusters of human organoid APs, chimpanzee organoid APs, endothelial cells (ECs), or iPSCs in G2M or G1 phases. We hierarchically clustered cells (Pearson correlation) using expression of genes that correlated with PC1 from PCA on human fetal cortex progenitor cells (*Camp et al., 2015*) and which are able to distinguish between cells in G2M and G1 phases. We selected the clusters with high or no expression and assigned them as G2/M or G1, respectively, and ignored the intermediate cells for SCDE. For the organoid APs, this assignment was consistent with an independent assignment using the method published by (*Scialdone et al., 2015*).

## Immunohistofluorescence

Cerebral organoids were fixed with 1% PFA in 120 mM phosphate buffer pH 7.4 for 20 min at room temperature and subjected to cryosectioning (14 µm) and immunofluorescence as described (*Camp et al., 2015*). The following primary antibodies were used: rabbit anti-PAX6 (PRB-278P; Covance), sheep anti-TBR2 (AF6166; R+D systems), rat anti-CTIP2 (ab18465; Abcam), rabbit anti-KI67 (ab15580; Abcam). The secondary antibodies, used in combination with DAPI staining, were all donkey-derived and conjugated with Alexa 488, 555 or 647 (Life Technologies). Images were acquired with a Zeiss LSM 880 Airy inverted microscope, using 10X (0.45 NA) and 20X (0.8 NA) Plan-Apochromat objectives, and analysed using Fiji. Quantifications were carried out in cortical regions of D28 and D52-54 cerebral organoids by counting, from the ventricular to the pial surface, either all PAX6 and TBR2 positive and negative nuclei stained by DAPI in 50 µm and 100 µm wide fields,

respectively, or all KI67-positive cells in 100 µm wide fields. An average of 350 cells per sample were counted. Statistical significance was calculated using the Mann–Whitney U-test.

## Cumulative EdU labeling

EdU was added to 52 day old cerebral organoids at a final concentration of 1 µg/ml (added from a 1 mg/ml EdU stock in PBS). The organoids were supplied with fresh medium containing EdU every six hours for up to 48 hr. Organoids were then collected in triplicates at the indicated time points (1, 2, 6, 24, 30/36, 48 hr) and processed as described above. For EdU detection, the Click-iT EdU Alexa Fluor 647 Imaging Kit (Invitrogen C10340) was used according to the manufacturer's instructions. Cell cycle parameters were determined using linear regression based on a model described previously (*Nowakowski et al., 1989*).

## Live imaging

Live tissue imaging was performed as described previously (*Mora-Bermudez et al., 2014*). In short, cerebral organoids or freshly dissected developing neocortex tissue were embedded in agarose (Sigma, Germany), sectioned with a vibratome (~200 µm, Leica, Germany), embedded in type Ia collagen (Cellmatrix, Japan), mounted in glass bottom microwell dishes (MatTek, Germany), and incubated with Hoechst 33342 (Sigma) as vital DNA dye. Tissue slices in the dish were further cultured for observation in a microscope stage incubation chamber (Pecon, Germany) kept at 37°C. iPSCs and B cells were likewise mounted in glass bottom microwell dishes previously coated for 1h with matrigel (BD Biosciecne) and poly-D-lysine (Sigma, Germany) respectively, and imaged under their respective standard culturing conditions (see above). Potential phototoxicity was stringently controlled as previously described (*Mora-Bermudez and Ellenberg, 2007*).

### Image analysis

Images were viewed and prepared with ImageJ (http://imagej.nih.gov/ij/). Brightness and contrast of images were recorded and adjusted linearly. Spindle orientation analysis was performed as described (*Mora-Bermudez et al., 2014*). In short, the degree values given in *Figure 4* are deviations from a perfect orthogonality with the local apical surface plane, as seen from a coronal perspective (*Figure 4A–F*). For *Figure 4G*, the maximal range of orientations per every mitotic AP was calculated from the formation of a metaphase plate to anaphase onset.

### Mitotic phase length determination

To measure the duration of mitotic phases, the start of each different phase was defined as follows, based on morphology, dynamics and condensation of chromosomes as reported by vital DNA staining (*Figures 5* and *6*). Prophase: the beginning of mitotic chromosome condensation; prometaphase + metaphase: the beginning of chromosome congression and alignment; anaphase: the beginning of chromosome segregation toward the mitotic poles of the dividing cell; telophase: the beginning of chromosome decondensation after maximal chromosome condensation in late anaphase and until a level indistinguishable from interphase was achieved. The total duration of mitosis was the sum of these phases. We note that our measurements of mitotic phases are limited by the use of chromosomes as markers. Nevertheless, the use of a single fluorescence channel allowed a very high time resolution (~1.1 min) for close monitoring of key chromosomal dynamics to delimit mitotic phases. Towards distinguishing between prometaphase and metaphase, we subdivided prometaphase + metaphase into 'prometaphase', defined here as the time in which chromosomes are congressing and aligning toward the formation of a metaphase plate, and 'metaphase', defined here as the time after every chromosome has been incorporated into a tight metaphase plate at the equatorial plane of the cell, and until anaphase onset.

### Statistical analysis

Data were tabulated with Excel (Microsoft, Redmond, WA) and analysed with GraphPad Prism (La Jolla, CA). Statistical tests: for two groups of observations, the Mann–Whitney U-test was used. For three or more groups, the Kruskal–Wallis ANOVA in conjunction with Dunn's Multiple Comparison test for pair-wise comparisons was used. Results were interpreted as statistically significant when $p < 0.05$.

## Acknowledgements

We thank the Services and Facilities of the Max Planck Institute of Molecular Cell Biology and Genetics for outstanding support, notably Jussi Helppi and his team of the Animal Facility, and Jan Peychl and his team of the Light Microscopy Facility. We thank David Andrijevic and Anne Weigert for help with maintenance and characterization of iPSC lines. We thank Marta Florio for assistance with human tissue dissection. We thank Andrea Musacchio and members of the Huttner, Treutlein and Pääbo labs for helpful discussions. We thank Fred Gage and Rick Livesey for kindly donating the PR818-5 iPSC line. SK was supported by a PhD fellowship of the Boehringer Ingelheim Fonds. SP was supported by the Paul G. Allen Family Foundation. WBH was supported by grants from the Deutsche Forschungsgemeinschaft (DFG, SFB 655, A2) and the European Research Council (ERC, 250197), by the DFG-funded Center for Regenerative Therapies Dresden, and by the Fonds der Chemischen Industrie. SP, BT and WBH were supported by the Max Planck Society.

## Additional information

### Funding

| Funder | Grant reference number | Author |
| --- | --- | --- |
| Boehringer Ingelheim Fonds | | Sabina Kanton |
| Paul G. Allen Family Foundation | | Svante Pääbo |
| Deutsche Forschungsgemeinschaft | DFG, SFB 655, A2 | Wieland B Huttner |
| European Research Council | ERC, 250197 | Wieland B Huttner |
| Deutsche Forschungsgemeinschaft | | Wieland B Huttner |
| Fonds der Chemischen Industrie | | Wieland B Huttner |
| Max Planck Society | | Svante Pääbo<br>Barbara Treutlein<br>Wieland B Huttner |

The funders had no role in study design, data collection and interpretation, or the decision to submit the work for publication.

### Author contributions

FM-B, Conceived the study, Designed the experiments, Performed and analysed live imaging experiments, Wrote the paper; FB, Conceived the study, Designed the experiments, Grew cerebral organoids, Performed and analysed organoid immunohistochemistry and cumulative EdU labelling, Wrote the paper; SK, Conceived the study, Designed the experiments, Grew cerebral organoids, Performed single-cell RNA-seq experiments, Analysed single-cell RNA-seq data, Wrote the paper; JGC, Conceived the study, Designed the experiments, Performed single-cell RNA-seq experiments, Analysed single-cell RNA-seq data, Wrote the paper; BVe, Analysed single-cell RNA-seq data, Provided information relevant for the interpretation of the data; KK, BVo, KO, TM, Prepared chimpanzee iPSC line Sandra A and orangutan iPSC line Toba, Provided information relevant for the interpretation of the data; ZH, Constructed human-chimpanzee consensus genome ; RL, Provided human fetal tissue, Provided information relevant for the interpretation of the data; SP, WBH, Conceived the study, Designed the experiments, Provided intellectual guidance in the interpretation of the data, Wrote the paper; BT, Conceived the study, Designed the experiments, Analysed single-cell RNA-seq data, Provided intellectual guidance in the interpretation of the data, Wrote the paper

### Author ORCIDs

Wieland B Huttner, http://orcid.org/0000-0003-4143-7201

## Ethics

Human subjects: Human fetal brain tissue (11-13 weeks post conception (wpc)) was obtained with informed written maternal consent followed by elective pregnancy termination. Research involving human tissue was approved by the Ethical Review Committee of the Universitaetsklinikum Carl Gustav Carus of the Technische Universitaet Dresden. In addition, research was approved by the Institutional Review Board of the Max Planck Institute of Molecular Cell Biology and Genetics.

Animal experimentation: Mice were kept pathogen-free at the Biomedical Services Facility of the MPI-CBG. All experiments using mice were performed according to the German Animal Welfare Legislation. In addition, research was approved by the Institutional Review Board of the Max Planck Institute of Molecular Cell Biology and Genetics.

## Additional files

### Major datasets

The following dataset was generated:

| Author(s) | Year | Dataset title | Dataset URL | Database, license, and accessibility information |
|---|---|---|---|---|
| Kanton S, Camp G, Treulein B | 2016 | Differences and similarities between human and chimpanzee neural progenitors during cerebral cortex development | https://www.ncbi.nlm.nih.gov/geo/query/acc.cgi?acc=GSE86207 | Publicly available at NCBI Gene Expression Omnibus (accession no: GSE86207) |

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
