## [Decision Letter]

Thank you for submitting your article "Differences and similarities between human and chimpanzee neural progenitors during cerebral cortex development" for consideration by *eLife*. Your article has been reviewed by three peer reviewers, and the evaluation has been overseen by a Reviewing Editor and K VijayRaghavan as the Senior Editor. The following individuals involved in review of your submission have agreed to reveal their identity: Victor Borrell (Reviewer #3).

The reviewers have discussed the reviews with one another and the Reviewing Editor has drafted this decision to help you prepare a revised submission.

Summary:

Mora-Bermúdez and colleagues address the fundamental question of what distinguishes human and chimpanzee brain stem progenitor cells leading to the dramatic difference in brain size between these species. To address this technologically very challenging question, the authors use state of the art approaches including brain organoid cultures and single cell RNAseq. In a very remarkable methodological tour-de-force, the authors develop brain organoids from human, chimpanzee and orangutan iPSCs, and go on to perform high resolution live imaging of individual progenitor cells. The authors find that same classes of neural stem progenitor cells in humans and chimps are nearly identical at the transcriptome level, and also in their behavior in the cell cycle. A consistent difference identified by the authors is a lengthening of 5 min of metaphase during cell division. The study is timely and addresses a very important question. The reviewers, however, identified a number of technical and conceptual issues that will require the authors' attention.

Essential revisions:

Reviewer 1's main criticism is conceptual. The reviewer argues that an important part of the regulation of cell cycle progression is post-translational. As the cell cycle differences were detected in the VZ (in AP) which is not the major proliferative region in the human and in chimpanzee cortex this finding may not necessarily provide sufficient explanation to the difference between the human and chimpanzee brains. If this is the main goal of this manuscript, it may be more informative to put more emphasis on the human and chimpanzee PAX6 TBR2 NSPCs / PAX6 TBR2- NSPCs, which were not analyzed due to their low numbers. Alternatively, it may be advisable to analyze the changes at the level of protein expression in the AP during metaphase. In case such a study is beyond the scope of the current paper, the manuscript may be modified emphasizing the validity of the organoid system to study the conservation and diversification of cortical development in different species, the technical advances conducted and how they can assist in approaching such a complex and difficult study.

Reviewer 2 writes that the authors begin by saying: "Notably, whereas no significant differences were observed at D28, at D52-D54, the proportion of PAX6 TBR2 NSPCs in the chimpanzee organoids was nearly twice that in the human organoids, and the proportion of PAX6 TBR2- NSPCs was correspondingly lower, whereas no significant difference between human and chimpanzee was observed for PAX6- TBR2+ NSPCs (Figure 2)." The reviewer wonders:

A) How many organoids were used to make the comparison?

B) More details are needed to understand the variation within each specie.

C) The developmental time period for human and chimpanzee is different in vivo. How could the authors account for this age difference in vitro in organoids? Should the time in culture be normalized for gestation length?

D) What is the evidence that the organoid developmental clock parallels the in vivo clock and how reliably does that clock operate from culture to culture?

E) The error bars in Figure 2 should not be drawn in a manner that obscures the lower SD boundary-error extends in both directions.

F) Although the authors build their story based on Figure 2 there is not much difference in the number of Pax6 Tbr2- cells (considered as APs) at Day 52-54 between chimpanzee and human. Does this include cells from all the layers of cortex or just VZ/SVZ or is it not possible to distinguish these layers in the organoids? How many cells are counted per sample? In the figure legend, n=5 at day 28 whereas n=17 at D52-54, Does 'n' represent different organoid sections or different sections from same organoid? The difference in the number of Pax6 Tbr2 cells between chimpanzee and human is not evident in the immunostaining images in Figure 2. The author should provide a better view of the image marking some cells with co-localization of Tbr2 and Pax6. Also, they should comment on what kind of cells are Pax6 Tbr2 cells or comment that the cell type cannot be identified due to the unclear lamination of their organoids.

Reviewer 2 has the following additional major concerns:

1) Figure 1—figure supplement 2: Why are different cell populations age biased? Almost all cells from 45d are clustered as mesenchyme cells and all cells from 62d are clustered as hindbrain cells.

2) Please define clearly APs for Figure 4-6. Are those Pax6 Tbr2- cells? In Figure 4–Figure 6, no markers have been used to assign those cells as AP. Markers such as Pax6 and Tbr2 with reporter plasmids could be used to trace the cells. This kind of experiment would support the data presented in Figure 2. In Figure 7, the authors have tried to address this question but they have used developing mouse neocortex to distinguish proliferative and neurogenic APs. The dynamics of prometaphase-metaphase is very different in mouse as compared to human or chimpanzee, therefore, it is difficult to relate that the lengthening of prometaphase-metaphase characterizes proliferating NPSCs in humans.

3) In Figure 5, there seems to be considerable difference in prophase between mouse (24.5 min) and human or chimpanzee (~40 min). However, in Figure 5—figure supplement 1, no significant difference is shown. Moreover, the bar value for prophase and anaphase does not correlate with the values shown in the live images in Figure 5. Please clarify this difference in figures.

Furthermore, they found lengthening of the metaphase in human apical progenitors compared to chimp and no difference was observed in other phases during cell mitosis, but they didn't show whether the length of cell cycle for this kind of progenitors is different, which is also important for cell fate determination.

4) Figure 8: The method to cluster single cells into G1 and G2/M is neither described nor cited. Moreover, the authors didn't include cells in S phase of cell cycle. If the authors used the same set of genes as described in Camp, Badsha et al. 2015, they divided AP cells into S/G2/M and G1 clusters and not G1 and G2/M. Please explain the method and also explain why S phase is not included in this analysis.

5) In Figure 8, authors show many genes (~80 genes) that are highly expressed throughout the human AP cell cycle and were specific to APs. What is the difference in the list of genes highly expressed in AP cells in humans obtained in Figure 3 and in Figure 8? According to the data, it would mean that the prometaphase-metaphase duration could be prolonged by expressing any of these genes in chimpanzee organoid.

6) They have shown the time of mitosis difference for apical progenitors between human and chimp, but what about the basal progenitors as they also contribute substantially to neuron output.

7) Debra L. Silver has reported that prolonged mitosis of progenitors could lead to apoptosis or differentiation (2016, Neuron). Based on this finding one would draw the opposite conclusion.

Reviewer 3's criticism is also mainly conceptual. The reviewer argues that 5 min lengthening of metaphase is unlikely to be the only difference between human and chimp, as this would hardly explain differences in brain growth during embryonic development between these species. Is it possible that the influence attributed to the orientation of cleavage plane on the fate of mammalian cortex progenitor cells is overestimated, as recently proposed by Martínez-Martínez et al. (Nat Comm 2016) and others? Are human organoids larger (more cells) than chimpanzee after a similar period in culture? This simple piece of information may help understand if these organoids are indeed helpful to unravel differences between chimp and human cortex development, and if so where to look. Do human aRGs undergo more rounds of cell division than chimp? Is there any evidence in the transcriptomic analy￼￼ses of differences in genes controlling cell cycle re-entry?

Reviewer 3 also asks: Are the differences observed in length of cell cycle phases (and those not observed) due to human-chimp differences in brain development? Or rather they may be due to the transition between iPSC and organoid? The authors nicely compare metaphase in blood B cells from chimp and human and don't find differences. But then, will they find similar differences between human and chimp when comparing organoids from other tissues (i.e. gut organoids)?

Finally, the reviewer argues: The authors state that "each chimpanzee cell represents a cell state on a continuum from NPSCs to neurons based on gene expression signatures". Whereas one agrees on the concept of transcriptomic continuity across cell types in development, how does this concept fit with the canonical criterion of classifying cell types in the cerebral cortex in discrete groups, as is also done in this study? Although this is clearly not the main focus of the study, this type of classification analysis is quite used throughout the manuscript, and so the authors would do well in discussing this point.

---

## [Author Response]

Summary:

*Mora-Bermúdez and colleagues address the fundamental question of what distinguishes human and chimpanzee brain stem progenitor cells leading to the dramatic difference in brain size between these species. To address this technologically very challenging question, the authors use state of the art approaches including brain organoid cultures and single cell RNAseq. In a very remarkable methodological tour-de-force, the authors develop brain organoids from human, chimpanzee and orangutan iPSCs, and go on to perform high resolution live imaging of individual progenitor cells. The authors find that same classes of neural stem progenitor cells in humans and chimps are nearly identical at the transcriptome level, and also in their behavior in the cell cycle. A consistent difference identified by the authors is a lengthening of 5 min of metaphase during cell division. The study is timely and addresses a very important question. The reviewers, however, identified a number of technical and conceptual issues that will require the authors' attention.*

We sincerely thank all three reviewers for their constructive criticism and overall positive feedback, which has helped us a lot to improve our manuscript. We also would like to thank the Reviewing Editor for kindly preparing the summary and essence of the reviewers' comments.

In revising the manuscript, we have added two new sets of data.

First, quantification of the length of prometaphase-metaphase in human and chimpanzee apical progenitors (APs) of day 52 (D52) cerebral organoids (Figure 5—figure supplement 2). Interestingly, comparison with the day 30 (D30) cerebral organoid data that were already included in the original version of our manuscripts reveals that human AP metaphase is shorter at D52 than D30 and not any more different in length from D52 chimpanzee APs. As the proportion of proliferative divisions of human APs would be expected to decrease, and the proportion of differentiative divisions to increase, from D30 to D52, these data are consistent with the concept that a longer metaphase reflects a greater tendency for proliferative than differentiative AP divisions.

Second, determination of AP cell cycle parameters by cumulative EdU labeling (Figure 5—figure supplement 4). This reveals an only small difference in total cell cycle length between human and chimpanzee APs (≈6% shorter in chimp). Interestingly, however, S-phase in human APs is ≈ 5 hr longer than in chimpanzee APs. As a long S-phase has previously been shown to be a hallmark of proliferative (as opposed to neurogenic) AP divisions (Arai et al. 2011), these data again are consistent with the concept that human APs exhibit a greater tendency for proliferative than differentiative divisions than chimpanzee APs.

Essential revisions:

*Reviewer 1's main criticism is conceptual. The reviewer argues that an important part of the regulation of cell cycle progression is post-translational. As the cell cycle differences were detected in the VZ (in AP) which is not the major proliferative region in the human and in chimpanzee cortex this finding may not necessarily provide sufficient explanation to the difference between the human and chimpanzee brains. If this is the main goal of this manuscript, it may be more informative to put more emphasis on the human and chimpanzee PAX6 TBR2 NSPCs / PAX6 TBR2- NSPCs, which were not analyzed due to their low numbers. Alternatively, it may be advisable to analyze the changes at the level of protein expression in the AP during metaphase. In case such a study is beyond the scope of the current paper, the manuscript may be modified emphasizing the validity of the organoid system to study the conservation and diversification of cortical development in different species, the technical advances conducted and how they can assist in approaching such a complex and difficult study.*

We agree with reviewer 1 that the VZ is not the major proliferative region in the human and chimpanzee cortex, which is the SVZ, in particular the OSVZ as first shown by Dehay, Kennedy and colleagues. Yet, differences in cell cycle parameters between human and chimpanzee apical progenitors (APs), especially at the relatively early stages of cortical development examined in the present study, could impact the extent of basal progenitor (BP) generation and hence SVZ formation. As shown in the Figure 5—figure supplement 4, we have now determined cell cycle parameters for human and chimpanzee PAX6 TBR2– APs using cumulative EdU labeling, and find a ≈3 hr difference in total cell cycle length between human and chimpanzee D52-54 organoid APs (human 46.5 h, chimpanzee 43.8 h). Interestingly, this difference is essentially accounted for by a longer S-phase of human APs (17.5 hr vs. 12.8 h), which has previously been shown to be a hallmark of proliferative (as opposed to neurogenic) AP divisions (Arai et al. 2011). The difference between human and chimpanzee APs with regard to the progression of mitosis (Figure 5) is in line with the longer S-phase finding, as the longer metaphase plate stage may similarly impact the mode of AP division and thus the fate of the AP progeny. These differences in cell cycle and mitosis parameters between human and chimpanzee APs are consistent with the anticipated differences in cortical development between the two species, as is now discussed in greater detail in the revised manuscript (Discussion, last paragraph).

We agree with reviewer 1 that it is worth emphasizing the validity of the organoid system to study cortical development in different primates, as well as the technical advancement, and have done so in the Introduction (last paragraph).

As to the issue of dissecting mitotic phases of TBR2 basal progenitors by live high-resolution time-lapse imaging: we could not analyze these cell divisions in numbers sufficient for quantification due to their lower abundance as compared to APs, as also pointed out by the reviewer. These cell populations were therefore only analyzed by tissue immunofluorescence and RNA-seq (Figure 1–Figure 2).

We agree that a proteomics analysis of mitotic APs would be very interesting, but – as implied by the reviewer – is beyond the scope of the present study.

Reviewer 2 writes that the authors begin by saying: "Notably, whereas no significant differences were observed at D28, at D52-D54, the proportion of PAX6 TBR2 NSPCs in the chimpanzee organoids was nearly twice that in the human organoids, and the proportion of PAX6 TBR2- NSPCs was correspondingly lower, whereas no significant difference between human and chimpanzee was observed for PAX6- TBR2 NSPCs (Figure 2)." The reviewer wonders:

*A) How many organoids were used to make the comparison?*

As mentioned in the legend to Figure 2, Figure 5 organoids per species were used at day 28 and 17 organoids per species were used at days 52-54.

*B) More details are needed to understand the variation within each specie.*

For sample numbers, please see response above. For further details, please see revised Methods.

*C) The developmental time period for human and chimpanzee is different* in vivo*. How could the authors account for this age difference* in vitro *in organoids? Should the time in culture be normalized for gestation length?*

Yes, the developmental time period in vivo is about 15% shorter for the chimpanzee as compared to humans. In line with this, we have preliminary data suggesting that cortical development in chimpanzee cerebral organoids proceeds slightly faster than in human cerebral organoids (unpublished data). However, these differences emerge only at later stages of organoid development and not at the stages included in the present study. Thus, the cortical regions compared between the two species were all positive for the deep-layer neuron marker Ctip2 (Figure 1) and negative for the upper-layer neuron marker Satb2 at D52-D54, and they were all Ctip2-negative at D28, indicating that the human and chimpanzee cerebral organoids were at a comparable stage in their development at these two time points. We therefore think that it would not be appropriate to normalize the time in culture for gestation length, at least not with regard to the relatively early stages of cortical development investigated in the present study.

*D) What is the evidence that the organoid developmental clock parallels the* in vivo *clock and how reliably does that clock operate from culture to culture?*

It is difficult to precisely determine whether the organoid developmental clock parallels the in vivo clock. Qualitatively, we and others (Lancaster et al. Nature 2013; Qian et al. Cell 2016; Pasca et al. Nature Methods 2015) have observed a general time-dependent progression of neurogenesis and transition to gliogenesis in human cerebral organoids prepared by various recently published protocols.

As to the issue how reliably that clock operates from culture to culture: We have analyzed human cerebral organoids from two independent iPSC lines and chimp cerebral organoids from two independent iPSC lines, and find that cortical development proceeds reproducibly from culture to culture.

E) The error bars in Figure 2 should not be drawn in a manner that obscures the lower SD boundary-error extends in both directions.

The figure has been revised such that the error bars now appear in both directions.

F) Although the authors build their story based on Figure 2 there is not much difference in the number of Pax6 Tbr2- cells (considered as APs) at Day 52-54 between chimpanzee and human. Does this include cells from all the layers of cortex or just VZ/SVZ or is it not possible to distinguish these layers in the organoids? How many cells are counted per sample? In the figure legend, n=5 at day 28 whereas n=17 at D52-54, Does 'n' represent different organoid sections or different sections from same organoid? The difference in the number of Pax6 Tbr2 cells between chimpanzee and human is not evident in the immunostaining images in Figure 2. The author should provide a better view of the image marking some cells with co-localization of Tbr2 and Pax6. Also, they should comment on what kind of cells are Pax6 Tbr2 cells or comment that the cell type cannot be identified due to the unclear lamination of their organoids.

The quantification of the Pax6 Tbr2– cells at day 52-54 (Figure 2), which yields a significantly lower value for chimpanzee than human, was performed across the entire cortical wall, i.e. from the ventricular to the pial surface, using 100-µm wide fields, as stated in the Methods section. As can be seen in Figure 2, the vast majority of these cells resided in the VZ (distinguishable as a germinal zone in the DAPI-stained images), consistent with them being APs.

An average of 350 cells per sample were counted. This information has now been added to the Methods section.

The ‘n’ represents different organoid sections.

As requested by the reviewer, Figure 2 has been revised and now includes representative insets indicating Pax6 Tbr2 cells at higher magnification, which illustrate the greater proportion of Pax6 Tbr2 cells in chimpanzee than human.

Based on their location in the SVZ and their marker expression, the Pax6 Tbr2 cells are likely to be basal progenitors committed to neuron production and endowed with self-renewal capacity.

*Reviewer 2 has the following additional major concerns:*

1) Figure 1—figure supplement 2: Why are different cell populations age biased? Almost all cells from 45d are clustered as mesenchyme cells and all cells from 62d are clustered as hindbrain cells.

We are aware that these two experiments (45d and 62d) appear as outlier populations. Based on our previous single cell transcriptome analysis of human organoids (Camp, Badsha et al. 2014 and unpublished observations), we do not think that this clustering pattern of chimpanzee cerebral organoid cells is related to the age of the respective organoid. Instead it likely reflects the relative composition of cells and patterned regions of those particular organoids, and our limited sampling of cells per organoid. The organoid protocol that we used (Lancaster et al. 2014) generates heterogeneous batches of organoids, where each organoid might have different brain regions. All organoids analyzed in this study contained one or more stratified cortical-like regions surrounding a ventricle. These cortical-like regions are often patterned as dorsal telencephalon (FOXG1 and OTX2-), however we have observed cortical regions that express ventral telencephalon or hindbrain markers (Camp, Badsha et al. 2015). For the 62d organoid, we observed one relatively large cortical-like region, which we microdissected and used for the scRNA-seq analysis. Based on the clustering pattern and marker gene expression, we concluded that cells from this 62d organoid were from a hindbrain-like region. In contrast, a different 61d organoid from the same batch contained many cerebral cortex cells, arguing against a developmental time-point phenomenon for the 62d organoid.

We have also observed an abundance of mesenchymal cells in some organoids, and have shown that these mesenchymal cells surround the periphery of cortical regions. In the case of the 45d organoid, we captured many of these mesenchymal cells and, unfortunately, no cerebral cortex cells.

It will be interesting to understand how the transcriptomes and genetic networks change as a function of age in human and chimpanzee cerebral organoids. However, we would need many more cells and time points to address this challenging question. Higher throughput scRNA-seq technologies, reporter lines marking correctly patterned cerebral cortex-like regions, and/or improved cerebral organoid protocols will help make this feasible in the future.

2) Please define clearly APs for Figure 4. Are those Pax6 Tbr2- cells? In Figure 4–Figure 6, no markers have been used to assign those cells as AP. Markers such as Pax6 and Tbr2 with reporter plasmids could be used to trace the cells. This kind of experiment would support the data presented in Figure 2. In Figure 7, the authors have tried to address this question but they have used developing mouse neocortex to distinguish proliferative and neurogenic APs. The dynamics of prometaphase-metaphase is very different in mouse as compared to human or chimpanzee, therefore, it is difficult to relate that the lengthening of prometaphase-metaphase characterizes proliferating NPSCs in humans.

The APs studied in Figure 4 and Figure 5 are indeed Pax6 Tbr2– cells, because in line with the data shown in Figure 2, virtually all mitotic figures at the ventricular surface are Pax6 Tbr2–. It is therefore not necessary to use Pax6 and Tbr2 reporter plasmids. Moreover, mitosis at the apical, ventricular surface, as is the case for the dividing cells analyzed in Figure 4 and Figure 5, is the defining criterion of an AP. We have now clarified this issue in the revised text (Introduction, second paragraph and subsection “Spindle orientation dynamics are similar in human and chimpanzee NSPCs”).

Although the absolute length of prometaphase-metaphase of mouse APs is shorter than that of hominid APs, the chromosome dynamics (e.g. their movements during congression, the building of the metaphase plate) are actually quite similar. Also, other mitotic phases (prophase, telophase, see Figure 5—figure supplement 1) are essentially the same for mouse and hominid APs. Therefore, we believe that it is legitimate to relate the longer prometaphase-metaphase of proliferating than neurogenic mouse APs to the longer prometaphase-metaphase of human than chimpanzee APs. Moreover, the shortening of metaphase of human APs with the progression of organoid cortical development from D30 to D52 (Figure 5—figure supplement 2), when proliferative AP divisions would be expected to decrease, is also consistent with the concept, derived from the dissection of mouse AP mitosis, that a longer metaphase reflects a greater tendency for proliferative than differentiative AP divisions. We have now further clarified this in the revised text (Results).

3) In Figure 5, there seems to be considerable difference in prophase between mouse (24.5 min) and human or chimpanzee (~40 min). However, in Figure 5—figure supplement 1, no significant difference is shown. Moreover, the bar value for prophase and anaphase does not correlate with the values shown in the live images in Figure 5. Please clarify this difference in figures.

There appears to be a misunderstanding. The time indicated on each image gives the time point relative to anaphase onset (0 min) when that image was taken, not the total time that the cell spent in that phase. Hence, the fact that the mouse prophase image was taken 24.5 min prior to anaphase onset whereas the hominid prophase images were taken 38-39 min prior to anaphase onset simply reflects the finding that prometaphase-metaphase is shorter in mouse APs than hominid APs. The length of prophase is essentially the same for mouse and hominid APs, as shown in Figure 5—figure supplement 1. To prevent such misunderstanding, we have now made this more clear in the figure legends.

*Furthermore, they found lengthening of the metaphase in human apical progenitors compared to chimp and no difference was observed in other phases during cell mitosis, but they didn't show whether the length of cell cycle for this kind of progenitors is different, which is also important for cell fate determination.*

As requested by the reviewer, we have now determined the length of the cell cycle for human and chimpanzee PAX6 TBR2– APs using cumulative EdU labeling, and find a ~2.7 hr difference between human and chimpanzee D52-54 organoid APs (human 46.5 h, chimpanzee 43.8 hr; Figure 5—figure supplement 4. While the reviewer is entirely correct in stating that the length of the cell cycle is, in principle, an important parameter for cell fate determination, we do not believe that in this particular case, the ~6% difference in total cell cycle length between human and chimpanzee APs will differentially impact cell fate. Rather, the longer S-phase of human APs (17.5 hr vs. 12.8 hr for the chimpanzee) is likely to be relevant, as this has previously been shown to be a hallmark of proliferative (as opposed to neurogenic) AP divisions (Arai et al. 2011).

4) Figure 8: The method to cluster single cells into G1 and G2/M is neither described nor cited. Moreover, the authors didn't include cells in S phase of cell cycle. If the authors used the same set of genes as described in Camp, Badsha et al. 2015, they divided AP cells into S/G2/M and G1 clusters and not G1 and G2/M. Please explain the method and also explain why S phase is not included in this analysis.

We used the same set of genes as described in Camp, Badsha et al. 2015 that divided AP cells into S/G2/M and G1 phases. Specifically we used the top 100 genes that correlated with PC1 from PCA on fetal cortex progenitor cells. We used these genes to hierarchically cluster human organoid APs, chimpanzee organoid APs, endothelial cells, or iPSCs. This generally gave two clear groups of cells that either expressed the genes highly or had low expression of these genes. There were additional sub-clusters of cells that partially expressed the genes. We selected the clusters with high or no/low expression and assigned them as G2/M or G1, respectively, and ignored the intermediate cells. This assignment was consistent with an unbiased assignment using the method published by (Scialdone et al. 2015). The cells that we see as intermediate or transiting cells were distributed between G2/M, S, and G1-phase, whereas the clusters we used in our analysis showed a clear assignment as G2/M or G1. For our differential expression analysis, we wanted to compare two groups of transcriptionally relatively homogeneous cells in G2/M and G1. We therefore excluded ambiguous cells and S-phase cells, which did not have a clear transcriptional signature in our experiment. We have included this information into the Methods section under the heading “Differential gene expression analysis” and added an additional supplemental figure (Figure 8—figure supplement 1).

5) In Figure 8, authors show many genes (~80 genes) that are highly expressed throughout the human AP cell cycle and were specific to APs. What is the difference in the list of genes highly expressed in AP cells in humans obtained in Figure 3 and in Figure 8? According to the data, it would mean that the prometaphase-metaphase duration could be prolonged by expressing any of these genes in chimpanzee organoid.

There are 10 genes that overlap between the genes highlighted in yellow in the top left quadrant of Figure 3 and the genes highlighted in Figure 8. These genes are APOLD1, BICC1, EFNB1, GSTM1, IFI44L, ITGB8, SDK2, SEMA5A, SLC35F1, ZNF516. The analysis in Figure 8 was performed to identify genes that are specifically expressed in G2/M APs independent of human-chimpanzee differential expression. Though we did not identify genes that are specific to G2/M APs, we did identify genes that are expressed throughout the cell cycle and generally enriched in APs compared to iPSCs and endothelial cells. The most differentially expressed genes between human and chimpanzee APs, which are also specific to APs, are listed above. It is possible that prometaphase-metaphase duration could be prolonged by over-expressing these genes in chimpanzee organoid. However, as the reviewers point out, the lengthening could also be regulated by post-transcriptional mechanisms and/or be affected by amino acids that have diverged between human and chimpanzee. Note that we have now specified these 10 genes in the text (Results, last paragraph).

*6) They have shown the time of mitosis difference for apical progenitors between human and chimp, but what about the basal progenitors as they also contribute substantially to neuron output.*

It would be very interesting indeed to compare the various mitotic phases of human and chimpanzee basal progenitors by live high-resolution time-lapse imaging. However, as also pointed out by reviewer 1, the abundance of basal progenitors in the cerebral organoids is much lower than that of APs, and the numbers of basal progenitors in mitosis were not sufficient for a reliable quantitative comparison.

*7) Debra L. Silver has reported that prolonged mitosis of progenitors could lead to apoptosis or differentiation (2016, Neuron). Based on this finding one would draw the opposite conclusion.*

Our results, linking the longer prometaphase-metaphase of human than chimpanzee APs to proliferative AP divisions, may at first sight appear to be opposite to those of Pilaz et al. 2016. However, we would like to emphasize that the experimental system employed by Pilaz et al. and that of the present study are very different from one another. The prometaphase-metaphase lengthening that we observed is a natural difference among three hominids and one rodent species, whereas Pilaz et al. generate a prolongation within one rodent species using drugs and genetic perturbations. The perturbations by Pilaz et al. cause very long prolongations of mitosis that may also be interpreted as mitotic arrest, which could also help explain the abundant cell death seen by Pilaz et al., as cell death is a hallmark of cells undergoing mitotic arrest. We have now clarified this issue in the revised text (Discussion).

*Reviewer 3's criticism is also mainly conceptual. The reviewer argues that 5 min lengthening of metaphase is unlikely to be the only difference between human and chimp, as this would hardly explain differences in brain growth during embryonic development between these species. Is it possible that the influence attributed to the orientation of cleavage plane on the fate of mammalian cortex progenitor cells is overestimated, as recently proposed by Martínez-Martínez et al. (Nat Comm 2016) and others? Are human organoids larger (more cells) than chimpanzee after a similar period in culture? This simple piece of information may help understand if these organoids are indeed helpful to unravel differences between chimp and human cortex development, and if so where to look. Do human aRGs undergo more rounds of cell division than chimp? Is there any evidence in the transcriptomic analyses of differences in genes controlling cell cycle re-entry?*

We agree with reviewer 3 that the 5-min lengthening of metaphase is unlikely to explain all of the difference between human and chimp in cerebral cortex growth during fetal development. The issue raised by the reviewer in this context, i.e. that a lack of a change in cleavage plane orientation does not preclude a change in daughter cell fate, is an excellent point. This possibility is indeed underscored by the recent work of Martínez-Martínez et al. (Nat Comm 2016) and also by work from the Matsuzaki lab that has shown that a perfectly vertical AP cleavage plane can give rise to either AP or BP daughter cells. Therefore, the reviewer is entirely correct in suggesting that factors other than cleavage plane orientation also influence AP daughter cell fate. In fact, in line with the reviewer's suggestion, our observation that there are no significant differences in spindle orientation between human and chimp made us analyze other aspects of AP division, which led to the finding of the metaphase lengthening. We have now discussed this issue in greater detail in the revised text (Discussion, last paragraph) and have added the Martínez-Martínez et al. 2016 reference.

As to the issue of human vs. chimpanzee cerebral organoid size:

The embryoid bodies formed after the iPS cell dissociation and plating may vary in size due to the initial differential clumping of cells. However, thereafter, there is no overt difference between the two species in the size of the organoids formed after a similar period in culture. In this context, we would like to point out that the lack of an overt difference in human vs. chimpanzee cerebral organoid size during the relatively early stages of organoid cortical development studied here does not necessarily imply that this system cannot be used to unravel differences between chimp and human cortex development, as these differences may only become apparent at later stages of organoid development.

As to the issue of human vs. chimpanzee organoid aRG rounds of cell division and cell cycle re-entry:

The ≈6% shorter total cell cycle length of chimpanzee APs as compared to human APs (Figure 5—figure supplement 4) does not support the possibility that human aRGs undergo more rounds of cell division than chimp. Also, there is no significantly greater abundance of cycling cells (Ki67 cells) in human as compared to chimpanzee cerebral organoid cortical regions (Figure 2), consistent with no major difference in NPC cell cycle re-entry between the two species.

*Reviewer 3 also asks: Are the differences observed in length of cell cycle phases (and those not observed) due to human-chimp differences in brain development? Or rather they may be due to the transition between iPSC and organoid? The authors nicely compare metaphase in blood B cells from chimp and human and don't find differences. But then, will they find similar differences between human and chimp when comparing organoids from other tissues (i.e. gut organoids)?*

Our data reveal that both types of differences are present. Compared to iPSCs, the length of prometaphase-metaphase is extended for both human and chimpanzee cerebral organoid APs (Figure 6). This shows that a general lengthening of prometaphase-metaphase is caused by the transition from iPSCs to cerebral organoids. However, the total length of prometaphase-metaphase is longer in human than in chimpanzee APs, indicating a species-specific difference (Figure 5). In addition, when prometaphase and metaphase were analyzed individually, our data show that, while both human and chimpanzee APs show a lengthening of prometaphase as compared to iPSCs, only the human APs showed also a lengthening of metaphase (Figure 6). This constitutes a species-specific difference. Taken together these results show that the longer prometaphase-metaphase in human APs is due to a lengthening not only of prometaphase, as seen in the chimpanzee, but also of metaphase. We have now made this more clear in the revised text (subsection “Gene expression in human and chimpanzee mitotic Aps”).

We agree with the reviewer that comparing organoids from other tissues would be interesting. However, this would require establishing other types of organoid systems in our lab to perform these complex and demanding experiments, and we think this would be out of the scope of the present study.

*Finally, the reviewer argues: The authors state that "each chimpanzee cell represents a cell state on a continuum from NPSCs to neurons based on gene expression signatures". Whereas one agrees on the concept of transcriptomic continuity across cell types in development, how does this concept fit with the canonical criterion of classifying cell types in the cerebral cortex in discrete groups, as is also done in this study? Although this is clearly not the main focus of the study, this type of classification analysis is quite used throughout the manuscript, and so the authors would do well in discussing this point.*

We agree with the reviewers that this is a very important point, and an interesting and challenging aspect of scRNA-seq data analysis, interpretation, and presentation. Traditionally, it has been convenient to classify cells into discrete types based on behavior, morphology and/or a small combination of marker genes in order to describe observations and make comparisons. To perform differential gene expression analysis using the Bayesian approach developed by Karchenko et al., which accounts for many of the assumptions inherent to single-cell transcriptome data, cell type classification is required. Therefore, we attempt to connect single cell transcriptome data with traditionally described cell types. In general, we do observe a continuum of transcriptome states across all cells, however we also find accumulations of cells that are similar with respect to the expression of groups of genes and steep transitions from such group of cells to other cells. These cells likely represent a cellular state that is relatively more stable than the apparent intermediates. The genes similarly expressed within such a cell group generally correlate well with previously described cell type marker genes and we therefore think we have the resolution to discretely classify APs, BPs, and neurons. Within each of these more stable cell states (=”cell types”), the continuity is less steep and we don’t have the resolution to determine if there is additional heterogeneity other than that related to cell cycle. There are of course some cells that are in between the previous and next state on the continuum. These intermediate cells are relatively less abundant but we force them into a discrete cell type. For example, if ~51% of a cell’s transcriptome resembles that of other BPs, and 49% of its transcriptome appears to be neuronal, then we assign this cell BP. Since these cells are relatively rare and present in both human and chimpanzee, we do not think that we introduce a bias into the differential gene expression analysis by including these cells.

We added the following statement to make this point more clear:

“Though this classification is convenient to describe the cell types present in the chimpanzee organoid, we note that many of the cells can be described as intermediates between APs, BPs, and different stages of neuron maturation.”